# Systemic iron availability differentially shapes tumor and brain iron handling in a sex-dependent manner in glioblastoma

Emily Tufano[1], Kondaiah Palsa[1], Rebecka O. Serpa[1,2,3], Timothy B. Helmuth[1], Gabriela Remit-Berthet[1], Sara Mills-Huffnagle[2,3], Mathias Kant[1], Aurosman Sahu[1], James R. Connor[1,2,3]*

1 Department of Neurosurgery, The Pennsylvania State University College of Medicine, Hershey, Pennsylvania, United States of America, 2 Department of Neuroscience and Experimental Therapeutics, The Pennsylvania State University College of Medicine, Hershey Pennsylvania, United States of America, 3 Penn State Neuroscience Institute, The Pennsylvania State University College of Medicine, Hershey, Pennsylvania, United States of America

* jconnor@pennstatehealth.psu.edu

## Abstract

Iron is essential for normal physiological function, yet dysregulation of iron metabolism is increasingly recognized as a hallmark of cancers such as glioblastoma (GBM). Recent clinical evidence suggests that systemic iron deficiency anemia (IDA) negatively impacts GBM outcomes in a sex-dependent manner, but the mechanisms linking systemic iron availability to tumor iron metabolism remain poorly understood. Here, we interrogate the impact of systemic iron through dietary modulation (control, iron deficiency (ID), and high iron diets), stratified by sex, on tumor iron handling and GBM outcomes utilizing an immune competent (C57BL/6) GBM (GL261) mouse model. Subsequently, we analyzed clinical samples to evaluate translational value. In the preclinical study, we show that iron deficiency decreased survival in males but conferred a slight survival advantage in females, consistent with prior clinical trends. Among circulating iron markers, only ferritin light chain (FTL), but not ferritin heavy chain (FTH) or serum iron, positively correlated with survival in males but not females. In the brain, contralateral iron levels reflected dietary iron status in males but not females, further supporting sex-dependent regulation of local and circulating iron. Notably, tumor iron content remained unchanged in males but was significantly elevated in ID female tumors, complemented by increased transferrin receptor (TfR1) and FTH expression. In clinical GBM samples, we observed non-statistically significant but similar survival trends across varying iron and ferritin levels, suggesting potential translational relevance of our exploratory model. These findings demonstrate that systemic iron availability exerts a sex-specific effect on tumor iron handling, highlighting a critical relationship between systemic and tumor iron regulation in GBM.

**Data availability statement:** All relevant data are within the manuscript and its Supporting Information files.

**Funding:** The following includes all funding received to support this study. National Center for Advancing Translational Sciences TL1-TR002016 (ET), and National Cancer Institute P01CA245705 (JRC). All funders had no role in study design, data collection and analysis, decision to publish, or preparation of the manuscript. There was no additional external funding received for this study.

**Competing interests:** The authors have declared that no competing interests exist.

## Introduction

Glioblastoma (GBM) is the most aggressive primary brain cancer, characterized by tumor heterogeneity, rapid proliferation, an immunosuppressive environment, and poor clinical outcomes [1,2]. Despite aggressive treatment with surgery, radiation and chemotherapy, the median survival remains around 14 months, with a local recurrence rate of 75–80% [3]. Resistance to therapy and devastating survival outcomes highlight the urgent need to identify new targets that may be exploited for therapeutic intervention [4]. Among these hallmarks of GBM, altered iron metabolism has emerged as a potential driver of tumor progression.

Iron is an essential element that is required for many physiological processes, including oxygen transport, DNA replication and repair, and neurotransmitter synthesis [5]. Cancer cells often demonstrate an increased demand for iron compared to normal cells, and this has been identified across a wide variety of cancers such as pancreatic, kidney, bladder, esophageal, liver, gastric, breast, colorectal, and brain cancer. In GBM, both increased iron import through transferrin receptor (TfR1) and increased iron delivery and storage through ferritin heavy chain (FTH) and ferritin light chain (FTL) have been observed [6]. Changes in iron handling across tumor and immune cell populations has been linked to increased reactive oxygen species (ROS), uncontrolled proliferation, and disrupted immune responses, thereby contributing to GBM progression [7]. Concurrent with these data, increased expression of iron proteins, including FTH and FTL, in the tumor are associated with reduced overall survival in patients, providing a strong correlative link between iron metabolism and tumor pathophysiology [6].

Recent studies have shown that systemic iron can also influence tumor biology, with both iron deficiency and overload being associated with poor outcomes across multiple cancer types [8,9]. Iron deficiency anemia (IDA) is the most common nutritional deficiency in the world and is also a common side effect of chemotherapy and cancer-related inflammation [10,11]. The consequences of IDA include reduced oxygen transport and immune competence, which may shape the tumor microenvironment to exacerbate growth [12]. In GBM, IDA has been linked to reduced overall patient survival in a sex-dependent manner, affecting only males [13]. Although iron overload is generally thought to promote oxidative stress and support tumor growth, plasma iron was positively correlated with survival [14]. In contrast, tumor iron did not correlate with survival outcomes, leaving the association between systemic and tumor iron unclear.

It is not yet known whether systemic iron availability can influence tumor iron pathways in vivo, or if there are measurable effects on tumor growth and survival. Additionally, the sex-bias reported in previous clinical data remains largely unexplored in translatable GBM models. In this study, we aim to address these gaps through preclinical experiments and clinical observations. Using an intracranial mouse tumor model, we evaluated the effects of dietary iron (control, iron deficient (ID), and high iron) on tumor growth and survival. This immune competent model recapitulates key features of GBM, and in our study this model was able to reproduce sex-dependent iron and ferritin patterns observed in patients, supporting its translational relevance

[15–17]. To complement these studies, we analyzed human GBM tumor and serum samples for circulating iron and ferritin levels.

After confirming IDA using hematologic parameters, we assessed tumor growth and survival following tumor cell injection. To distinguish systemic versus tumor-iron specific effects, we measured circulating serum iron, contralateral brain iron, and tumor iron. We further examined expression of key iron-related proteins including FTH, FTL, and TfR1, exploring sex differences across these measurements to understand how systemic iron responses may vary between males and females in GBM. To emphasize the clinical implications of altered iron homeostasis, we investigated iron and ferritin associations with survival in human GBM cohorts, separating our analyses by sex. Together, these studies provide new insights into how systemic iron availability, through dietary modulation, influences GBM progression and highlights the importance of differentiating the respective roles of systemic and tumoral iron in clinical outcomes.

## Materials & methods

### Diets & experimental procedure

A total of 50 C57BL/6 male (n = 26) and female (n = 24) were obtained to investigate the role of three varying dietary iron levels on tumor growth: Control, iron deficient, and high iron. Dietary iron deficiency protocols followed established methods. Mice were placed in group assignments by an individual not involved in downstream data collection. The groups were: Control (n = 35; 35 mg/kg Fe, AIN-93G, Inotiv, Cat. #TD.94045) and iron deficient (n = 17; 3.5 mg/kg Fe, #AIN-93, Inotiv, Cat. #TD.10210),) diets at postnatal day (PND) 21. All diets were formulated based on guidelines set by the American Institute of Nutrition (AIN), and all nutritional requirements, excluding the iron in the iron deficient diet group, were met [18]. Animals were maintained on diets until PND90, and blood samples were collected to evaluate iron deficiency by hemoglobin and hematocrit levels. Pre-determined exclusion criteria required that animals assigned to the IDA group exhibit hematological evidence of iron deficiency by PND 85. Animals failing to meet this threshold would have been excluded. No animals met the exclusion criteria. Immediately following confirmation of IDA compared to controls, animals underwent stereotaxic surgery for brain tumor implantation. At the time of tumor implantation, iron-deficient mice continued iron deficient diets, whereas control diet mice were allocated to either remain on control diets or transition to high iron diets. Diets were maintained for the remainder of the experiment (until the primary endpoint, about 17–37 days post injection), resulting in three experimental groups: control (female n = 8, male n = 9), iron deficient (female n = 8, male n = 9), and high iron (female n = 8, male n = 8; 350 mg/kg Fe, Global rodent diet, 2018 Teklad 18% Protein Rodent Diets, Cat. #NC0035505) diets. Animals were housed under the direct supervision of the Department of Comparative Medicine at Penn State College of Medicine in a standard temperature environment of 24 ± 1°C and a 12-hour light/dark cycle with lights on at 07:00h and off at 19:00h.

### Intracranial tumor implantation

Intracranial tumor implantations with the well-characterized mouse glioma GL261 syngeneic tumor cell lines were performed as previously described [19,20]. PND90 Male and female mice were deeply anesthetized with inhaled 5% isoflurane and given 0.5–1 mg/kg Buprenorphine SR subcutaneously for pain relief. Tumor cells were injected into the right hemisphere 1.5 mm posterior and 0.5 mm lateral to bregma at a depth of 3.5 mm from the cortical surface using a syringe mounted on a stereotaxic apparatus. Each injection contained 3μl of serum-free Dulbecco's modified eagle medium (DMEM) medium containing 20,000 GL261 cells. Directly following the procedure, mice were kept warm in a 37°C incubator single-housed until fully recovered. Mice were also observed for the premature onset of neurological symptoms, weight loss, or other behavioral indices of the humane primary endpoint (described below). All animal procedures adhered to institutional regulations and were performed in accordance with the Institutional Animal Care and Use Committee (IACUC) approved protocols at Penn State College of Medicine (Protocol 202101822).

## Hematology

Blood samples were collected the week of tumor implantation to evaluate hemoglobin and hematocrit levels to confirm iron deficiency or adequacy. 100 μL of blood was collected via the submandibular vein under 5% isoflurane anesthetic using anticoagulant Dipotassium EDTA to prevent clotting (RAM Scientific Safe-T-Fill Capillary Blood Collection, 14-915-50). Subjects' hemoglobin and hematocrit were analyzed using the Heska HT5 Veterinarian Hematology Analyzer (Heska Antech Company, Loveland, CO, USA).

## Magnetic resonance imaging (MRI)

All the image data were acquired on a 7T magnetic resonance imaging (MRI) scanner (Bruker BIOSPEC 70/20 USR) at the MRI Core Facility of College of Medicine. MRI scans were performed to assess tumor burden via T1- and T2-weighted sequences. For T1-weighted imaging (pre- and post-contrast), the acquisition parameters were as follows: Echo time of 8 ms, repetition time of 785 ms, with a plane resolution of 0.078 x 0.078 mm and a slice thickness of 0.5 mm. Animals received 0.0005 mL of Gadavist (gadobutrol) contrast diluted to a working concentration of 0.01 mg/kg. T2-weighted scans were acquired with an echo time of 35 ms and a repetition time of 3570 ms at the same resolution and slice thickness. The tumor regions were segmented and analyzed using he ITK snap tool. Investigators performing MRI acquisition and downstream analyses were blinded to diet group during data collection and analysis.

## Cell culture

Syngeneic mouse GBM cell line GL261s were grown in adherent conditions in DMEM high glucose media with 10% fetal bovine serum and 1% penicillin-streptomycin. Cells were maintained in humidified incubators at 37°C and 5% carbon dioxide and media was replaced every other day. Cells were passaged with trypsin and phosphate-buffered saline at 70–80% confluence.

## Measurement of Iron in serum, brain, and tumor tissue

Serum, tumor, and contralateral brain tissues were collected at the primary endpoint for the mice. Tumor and serum were collected from patients at the time of surgery, prior to treatment. Serum, brain and tumor tissues were digested overnight as previously described in 70% trace metal grade nitric acid at 60°C, followed by a subsequent dilution in MilliQ water [21]. Iron concentration was determined by inductively coupled atomic emission spectroscopy (ICP-AES, relative standard deviation < 2%) for mouse tumor (Limit of Detection (LOD) = 0.005 μg/ml) and ICP-mass spectrometry (ICP-MS) for human tumor (LOD = 0.18ng/mL), human serum (LOD = 0.18 ng/mL), mouse serum (LOD = 0.50ng/mL) and contralateral hemisphere samples (LOD = 0.33 ng/mL). No samples analyzed fell below the LOD. Results were expressed as g of Fe/g of tissue weight, and g of Fe/mL for serum volume. Investigators performing ICP measurements were blinded to diet group during data collection and analysis.

## Survival and end point

Tumor burden was assessed daily based on humane primary endpoint measures of significant body weight loss (~20%), body condition score < 2, and impaired mobility or behavioral changes. Upon reaching this endpoint, subjects were immediately anesthetized intraperitoneally with ketamine (100 mg/kg) and xylazine (10 mg/kg), followed by transcardiac perfusion with 0.9% saline. 9 animals reached the primary endpoint between monitoring periods and perfusion was not performed. These animals were excluded from further tumor and brain iron estimation analyses. Brain tissue was rapidly collected on ice, in which the tumor was dissected out and separated from the ipsilateral non-tumor tissue (if applicable) and contralateral brain tissue. Blood samples were collected with (described previously) and without the presence of an anti-coagulant, and serum was obtained through clotting the sample for approximately 30 minutes, followed by

centrifugation at 2000xg for 15 minutes at 4°C. Liver samples were also collected at this time, and all tissues and serum were stored at −80°C for subsequent analysis.

## Western blot

Western blot analysis was performed as described previously [22]. Tumor tissue homogenates were lysed in NP40 buffer (50mM Tris-HCL (pH 7.4), 150mM NaCl, 1% NP40, and 5mM EDTA) with protease inhibitor cocktail. Samples were homogenized manually, sonicated, and centrifuged at 12,000 rpm for 15 min at 4°C. Protein concentrations in the supernatant were determined using the Bradford protein determination assay. Equal amounts of protein were separated on 4–20% SDS-PAGE gels under reducing conditions and transferred to PVDF membranes. Membranes were blocked with 5% fat-free milk in TBST and incubated overnight at 4°C with the following primary antibodies: transferrin receptor/ CD71 (TfR1/CD71) (Santa Cruz Biotechnology; 1:250, SC-65882), ferritin light chain (FTL) (Abcam, 1:1000, ab69090) and β-actin (Sigma, 1:1000, A5441). After washing with TBST the following day, membranes were incubated with appropriate HRP-conjugated secondary antibody (1:5000, Cytiva). Protein bands were detected using ECL reagents (Santa Cruz Biotechnology, SC-2048) on an Amersham Imager 600 (GE Amersham). Band intensity was quantified using ImageJ software (NIH, Bethesda, MD, USA) and normalized to the corresponding loading control. Raw western blot images are provided in S5 Fig. to support the quantitative analyses performed.

## Enzyme-linked immunosorbent assay (ELISA)

Mouse serum and tumor ferritin heavy chain (FTH) and ferritin light chain (FTL) enzyme-linked immunosorbent assays (ELISAs) were performed according to manufacturer's protocols (FTH: Life Span Biosciences, USA, # LS-F7520-1, limit of Detection (LOD) = 6.1pg/mL, intra-assay CV < 10% and inter-assay coefficient of variation (CV)<12%; FTL: Life Span Biosciences, USA, # LS-F8829, LOD = 0.056ng/mL, intra-assay CV < 10% and inter-assay CV < 12%). Serum samples were diluted 1:500 for both FTH and FTL measurements. One standard diet female sample fell below the LOD for serum FTH. One iron deficient male and one control male sample fell below the LOD for serum FTL. Samples below the LOD were excluded from analysis. For tumor, equal protein concentrations were determined using the Bradford protein determination assay, and only 1 iron deficient male fell below the LOD and was excluded from analysis. Human serum and tumor FTH and FTL ELISAs were performed according to manufacturer's protocols (FTH: Life Span Biosciences, USA, # LS-F22867, LOD = 7.5 ng/mL, intra-assay CV < 6.42% and inter-assay CV < 7.34%; FTL: Life Span Biosciences, USA, # LS-F6558, LOD = 0.16ng/mL, intra-assay CV < 10% and inter-assay CV < 12%). Serum samples were diluted 1:500 for both FTH and FTL measurements. One serum sample fell below the LOD for FTH and was excluded from analysis. For tumor samples, equal protein concentrations were determined using the Bradford protein determination assay (Thermo Fisher Scientific, Cat. #23200) and 10 patients were excluded who fell below the LOD.

## Collection of clinical plasma and tumor samples

Samples from Penn State Neuroscience Institute's Biorepository STUDY00002914 are from subjects who have given permission for the use of their de-identified samples for future research. Plasma and tumor samples were collected from 70 consented GBM patients (28 females and 42 males), 51 of which had matched samples. Plasma was obtained prior to tumor resection and subsequent treatments. Resected tumor tissue was collected during the procedure and immediately stored at −80°C together with the matched plasma until batch processing. The full de-identified dataset was accessed for research purposes on 08/28/2025. Tumor tissue was visually inspected for the removal of regions with overt necrosis or hemorrhage to ensure consistency between samples. Clinical information for each patient was extracted from the electronic medical record by a neutral system (honest broker) in accordance with the Biorepository protocol to further protect privacy by separating patient identifiers from samples and results.

Kaplan-Meier survival curves were generated for several measurements in the plasma and tumor. For all survival analyses, patients were stratified into high and low groups based on the sex-specific medians for each measurement (tumor iron, tumor FTH, serum iron, serum FTL, serum FTL). Multivariate cox proportional hazards regression was performed, adjusting for relevant clinical covariates such as patient sex, age at diagnosis, white blood cell count, extent of tumor resection, receipt of chemotherapy, dexamethasone, and/or radiation. Kaplan-Meier curves prior to multivariate cox proportional hazards regression can be found in S2-S4 Figs.

### Statistical analysis plan

All statistical analyses for preclinical data can be found in S1 Table, and all analyses for clinical data can be found in S2 Table.

**Preclinical data.** Statistical analysis was performed using Prism software version 10.4.1 (GraphPad Software, LLC., San Diego, CA, USA). Data results were expressed as mean ± standard deviation (SD). Outlier tests were performed when necessary, using the robust regression outlier (ROUT) method at Q = 1%. Two-way ANOVA, for sex and diet as factors, with Tukey post-hoc analysis (three groups), one-way ANOVA with Dunnett's multiple comparison correction (three groups), or unpaired t-test (two groups) were used to evaluate statistical differences, and a p-value < 0.05 was considered significant.

**Clinical data.** Kaplan-Meier survival curves were generated for several measurements in the plasma and tumor. For all survival analyses, patients were stratified into high and low groups based on the sex-specific medians for each measurement (tumor iron, serum iron, tumor FTH, serum FTL, serum FTL). Survival distributions between these groups were assessed for any relationships with patient outcomes. Multivariate cox proportional hazards regression was performed, adjusting for relevant clinical covariates such as patient sex, age at diagnosis, and receipt of chemotherapy and/or radiation.

## Results

### Physiological evaluation of iron deficiency anemia (IDA) and control mice

To investigate the influence of systemic iron status on GBM progression, hemoglobin and hematocrit were measured at PND90 to verify iron status prior to tumor implantation. Following tumor injection, mice remained on control, ID, or high iron diets for the duration of the study as outlined in Fig 1. IDA males had significantly lower hemoglobin (11.07 ± 1.122

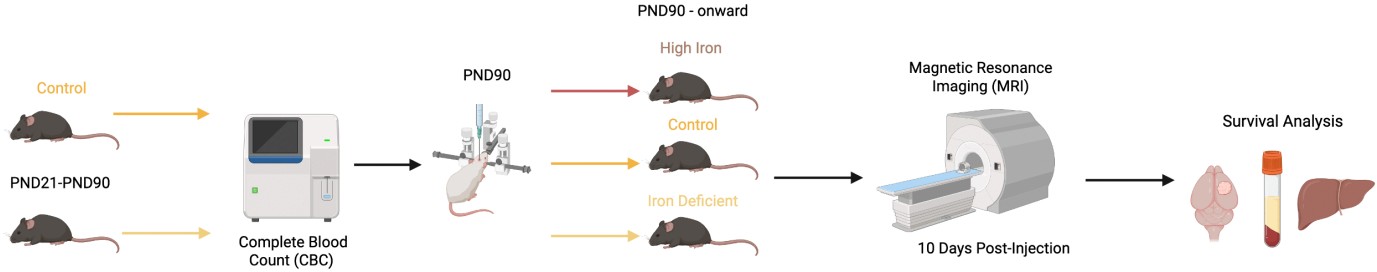

**Fig 1. Experimental design for dietary manipulation and tumor inoculation.** Beginning postnatal day (PND) 21, mice were assigned to either a control diet 35 mg/kg Fe, AIN-93G, n = 33) or an iron deficient (ID) diet 3.5 mg/kg Fe, #AIN-93, n = 17). On PND90, iron deficiency anemia (IDA) was confirmed in the ID groups using hematological measures. After confirmation, control diet animals were assigned to either remain on the control diet (n = 17) or switched to a high iron diet (350 mg/kg Fe, #NC0035505, n = 16) for the remainder of the study following tumor implantation. IDA mice remained on the ID diet throughout the experiment (n = 17). Tumor volume was assessed using magnetic resonance imaging (MRI) 10 days after tumor injection. Animals were monitored daily until experimental endpoint, at which time serum, liver tissue, whole tumors, and contralateral brain tissue were collected for downstream analysis.

vs 14.42±0.3552 g/dL, p<0.0001) and hematocrit (36.8±3.441 vs 45.66±1.861%, p<0.0001) relative to controls (Figs 2A-B). Similarly, IDA females exhibited reduced hemoglobin (11.48±1.543 vs 14.52±0.7029 g/dL, p<0.0001) and hematocrit (37.09±4.7887 vs 45.33±1.966%, p<0.0001) relative to controls (Figs 2C-D). Hemoglobin and hematocrit were also measured at the humane endpoint. Interestingly, IDA females had significantly lower hemoglobin compared to control mice (11.0±2.107 vs 15.15±1.401), while IDA males did not (13.26±1.394 vs 13.86±2.553) (S1 A,C Fig). This relationship also held true for hematocrit, where IDA females were significantly lower than controls (34.23±5.060 vs 47.42±3.316) and IDA males were not (39.24±4.315 vs 40.92±6.219) (S1 B,D Fig).

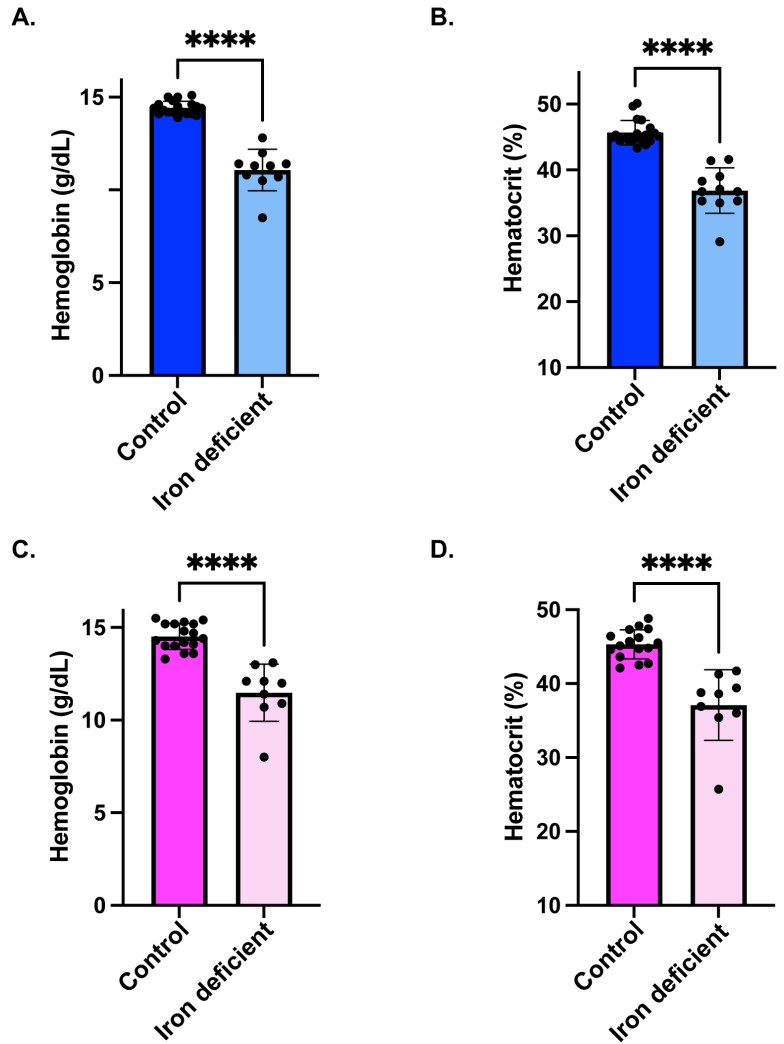

**Fig 2. Iron deficiency anemia confirmation via hemoglobin and hematocrit measures. A-B)** Mean hemoglobin and hematocrit measures in males (±SD): By postnatal day (PND) 90, males on the iron deficient (ID) diet exhibited significantly reduced hemoglobin (****p<0.0001) and hematocrit (****p<0.0001) (n=19 Control, n=10 IDA), confirming iron deficiency anemia (IDA). **C-D)** Mean hemoglobin and hematocrit measures in females (±SD): By postnatal day (PND) 90, females on the ID diet exhibited significantly reduced hemoglobin (****p<0.0001) and hematocrit (****p<0.0001), confirming IDA (n=17 Control, n=9 IDA).

## Dietary iron influences GBM tumor growth and survival in a sex-dependent manner

Following confirmation of IDA in adult mice, we next evaluated tumor volume at 10 days post-implantation and tracked survival to assess the consequence dietary iron status on GBM progression. Male mice on ID diets showed a 234% increase in tumor growth compared to controls ($1.867 \pm 1.711$ mm$^3$ vs. $0.4425 \pm 0.9117$), although this difference did not reach statistical significance ($p = 0.06$) (Fig 3A). Tumor volume was also modestly increased in males on the high iron diet ($1.396 \pm 0.5083$ mm$^3$ vs. $0.4425 \pm 0.9117$, $p = 0.28$). In contrast, female mice showed the opposite

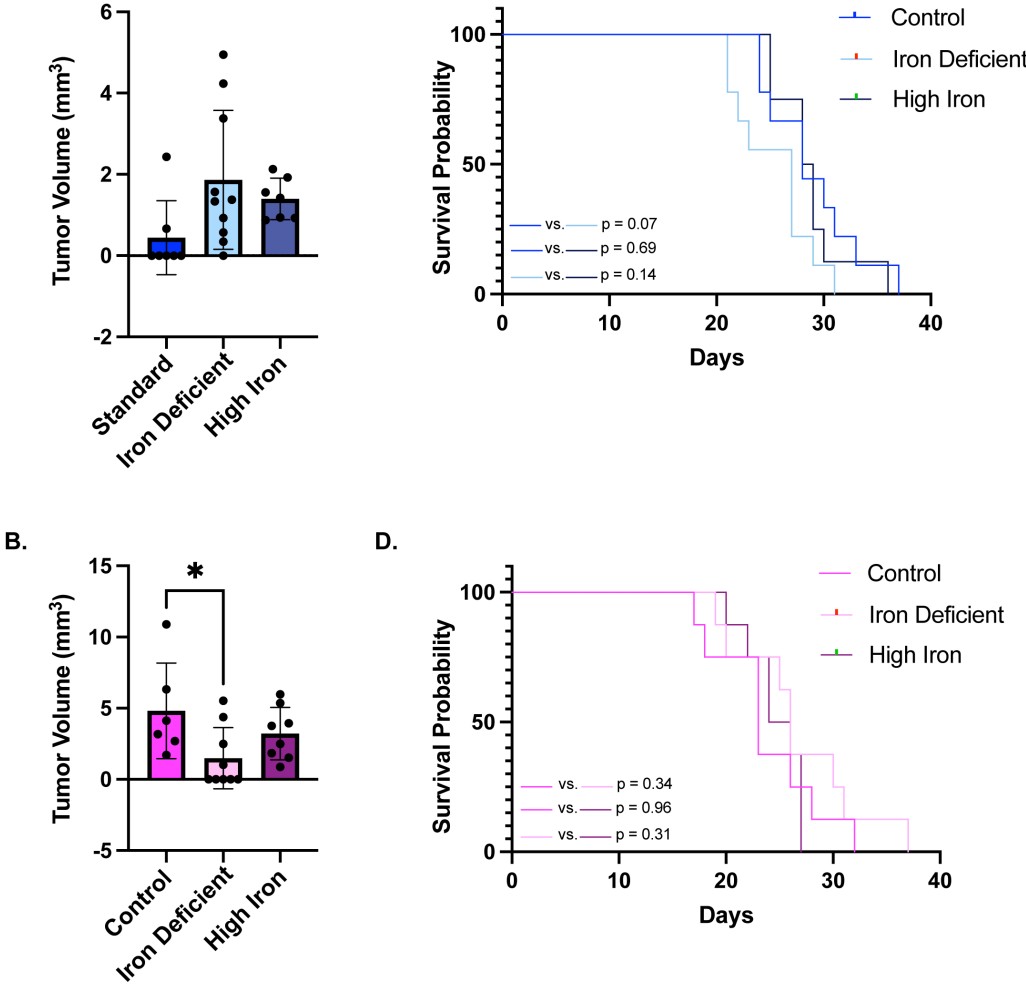

**Fig 3. Tumor volume and survival. A)** Mean tumor volume evaluated at 10 days post-injection by magnetic resonance imaging in males ($\pm$SD): There were no significant differences in tumor volume between controls (n=7) and iron deficient (ID) (n=10) ($p=0.06$) or high iron groups (n=7) ($p=0.28$). **B)** Mean tumor volume evaluated at 10 days post-injection in females ($\pm$SD): There was a significant reduction in tumor volume in ID females (n=6) (*$p=0.03$) and no significant difference in high iron females (n=8) ($p=0.37$) versus control females (n=6). All tumor volumes were evaluated for statistical significance using a one-way ANOVA with a Dunnett's multiple comparisons test. **C)** Kaplan-Meier survival analysis of males on control, ID, and high iron diets: There were no significant differences in survival between control and IDA ($p=0.07$), control and high iron ($p=0.69$) or high iron and IDA males ($p=0.14$). **D)** Kaplan-Meier survival analysis of females on control, ID, and high iron diets: There were no significant differences in survival between control and IDA ($p=0.34$), control and high iron ($p=0.96$) or high iron and IDA females ($p=0.31$). Data were evaluated for statistical significance using a Log-rank (Mantel-Cox) test.

trend: IDA significantly reduced tumor volume compared to controls (1.492±2.150 vs 4.82±3.361 mm³, p=0.03), and high iron also slightly reduced tumor volume (3.218±1.843 vs 4.82±3.361 mm³), though not significantly (p=0.37) (Fig 3B).

Survival analysis using Kaplan-Meier curves with log-rank tests further revealed sex-specific outcomes aligned with early tumor volume patterns. In males, ID diets were associated with reduced survival compared to controls, though this did not reach statistical significance (27 vs. 28 days, p=0.07) (Fig 3C). Survival in the high-iron group was similar to controls (28.5 vs 28 days, p=0.69). In females, ID diets conferred a slight survival advantage relative to controls (26 vs 23 days, p=0.34) consistent with reduced tumor volumes (Fig 3D). Similarly, high iron females showed no significant difference in survival compared to controls (25 vs 23 days, p=0.96).

## Contralateral Hemisphere iron concentration following dietary manipulation is sex-dependent

Given the sex-dependent differences in tumor growth and survival observed in the varying dietary conditions, we sought to determine differences in brain iron regulation outside of the tumor microenvironment. To explore the impact of systemic versus brain iron on GBM outcomes, we measured iron content in the contralateral hemisphere of male and female mice in each diet group. IDA males had significantly reduced iron in the contralateral hemisphere compared to control (5.300±0.46 vs 7.743±3.09 g/tissue weight, p=0.01) and high iron males (5.300±0.46 vs 8.173±2.87 g/tissue weight, p=0.009), whereas no differences were observed between control male and high iron males (p=0.67) or among any female groups (Fig 4A). Despite the lack of differences in female contralateral hemisphere iron between groups, high iron significantly correlated to improved survival across all cohorts (R=0.4192, p=0.002), and when analyzed by sex (males: R=0.595, p=0.002; females: R=0.3442, p=0.05) (Figs 4B-D). Interestingly, serum iron did not significantly correlate with contralateral iron across all mice (R=0.2115, p=0.10) or in females alone (R=−0.0507, p=0.42) (Figs 4E and G). However, a positive association was found between brain iron content and serum iron in males (R=0.5329, p=0.01) (Fig 4F). Together, these data suggest that contralateral hemispheric iron content, rather than serum iron alone, may influence GBM outcomes in a sex-specific manner.

## Sex-dependent tumor iron content is associated with survival

We next examined tumor iron levels and their association with GBM outcomes in the context of sex-dependent brain iron regulation. Comparing tumor iron between diets further highlighted sex-specific effects. In males, tumor iron did not differ significantly between control and IDA (18.14±7.286 vs 16.68±8.673 g/tissue weight, p=0.90) or high iron diets (18.14±7.286 vs 17.64±8.179 g/tissue weight, p=0.99) (Fig 5C). In females, however, IDA mice had significantly elevated tumor iron levels compared to control (17.34±2.563 vs 10.19±3.469 g/tissue weight, p=0.02) (Fig 5D). High iron female mice also had slightly elevated tumor iron compared to control (15.60±5.757 vs 10.19±3.469 g/tissue weight), though this failed to reach statistical significance (p=0.07).

Survival analysis revealed no significant correlation between tumor iron and survival when assessing all cohorts (R=0.2539, p=0.10) or in females (R=−0.02, p=0.46), but males alone had a significant positive association between these parameters (R=0.4260, p=0.02) (Figs 5E-F). Comparatively, Kaplan-Meier analysis showed no survival differences based on high versus low tumor iron in the combined cohort (27 vs 26 days, p=0.10) or in females alone (24.5 vs 26 days, p=0.88) (Figs 5B and H). In contrast, a survival advantage was observed for males with high tumor iron compared to low tumor iron (29.5 vs 27 days, p=0.02) (Fig 5G). When examining clinical data, there is a modest separation in survival curves observed in males, with higher tumor iron associated with slightly longer survival (HR=0.56 (0.16–1.90), p=0.35), whereas no comparable separation was evident in females (HR=0.90 (0.29–2.77), p=0.85) or in the combined cohort (HR=0.98 (0.55–1.77), p=0.96) (Figs 5I-K). Despite the lack of statistical significance in both males and females, the directionality of this pattern was consistent with those observed in the preclinical model.

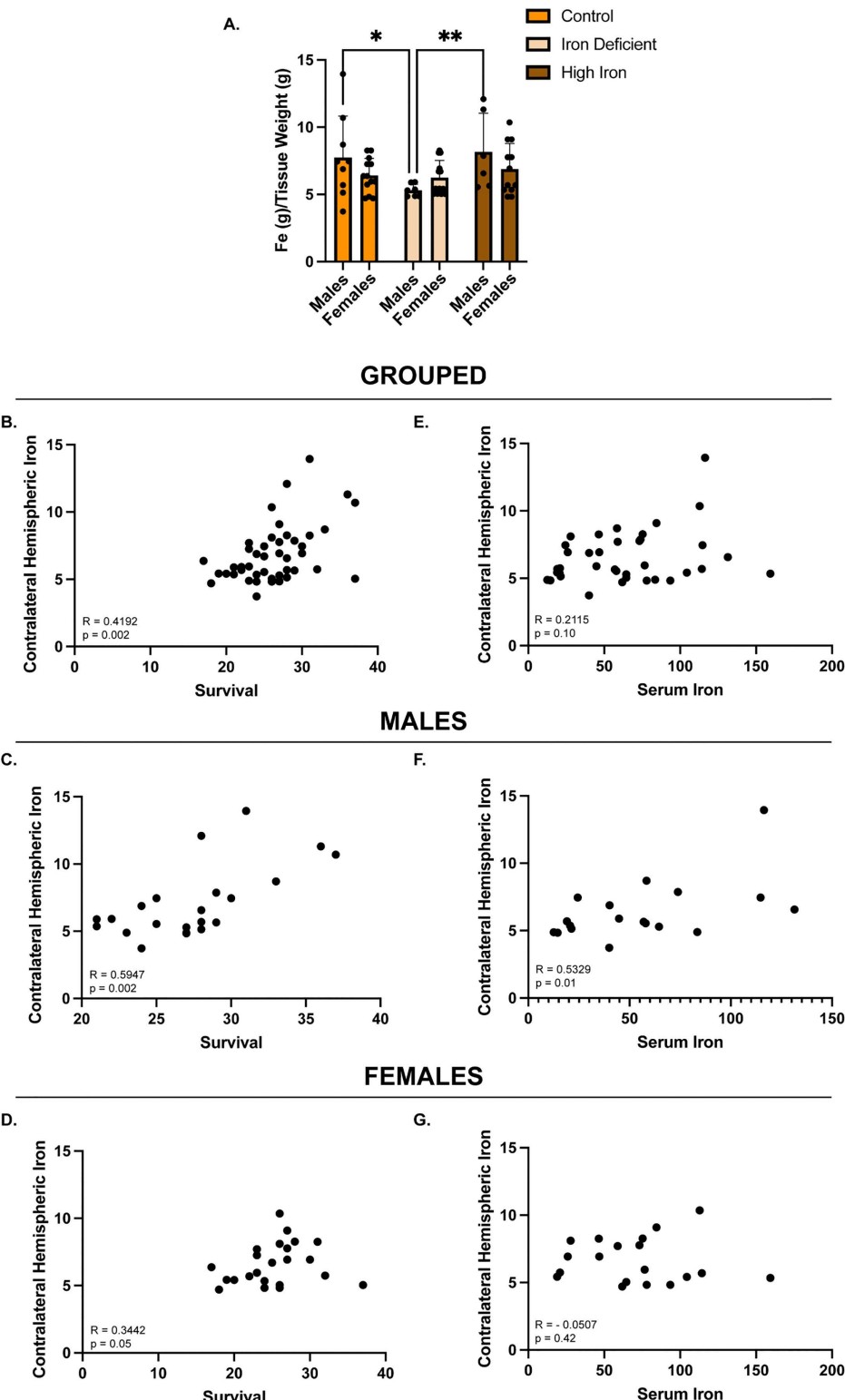

**Fig 4. Contralateral hemisphere iron concentration. A)** Mean iron concentration levels in male and female contralateral hemisphere across dietary groups: In males, contralateral hemispheric iron levels differed significantly between the control (n = 9) and iron deficiency anemia (IDA) groups (n = 7) (*p = 0.01) and IDA versus high iron groups (n = 6) (**p = 0.009). No significant differences were observed among any of the female dietary groups (Control n = 13; IDA n = 13; High Iron n = 12). Contralateral hemisphere iron data were evaluated for statistical significance using a two-way ANOVA.

**B-D)** Contralateral hemisphere iron correlations to survival as a grouped cohort and in males and female separately: There was a significant correlation amongst the combined cohort (n = 45) (**p = 0.002), males (n = 22) (**p = 0.002), and females (n = 23) (*p = 0.05). Correlations were evaluated for statistical significance using the Spearman r coefficient of correlation. **E-G)** Correlations between contralateral hemispheric iron and serum iron for the combined cohort and for males and females analyzed separately: No significant correlation was observed in the combined cohort (n = 37) (p = 0.10) or in females (n = 19) (p = 0.42). There was a significant positive correlation between contralateral hemispheric iron and serum iron in males (n = 18) (**p = 0.01).

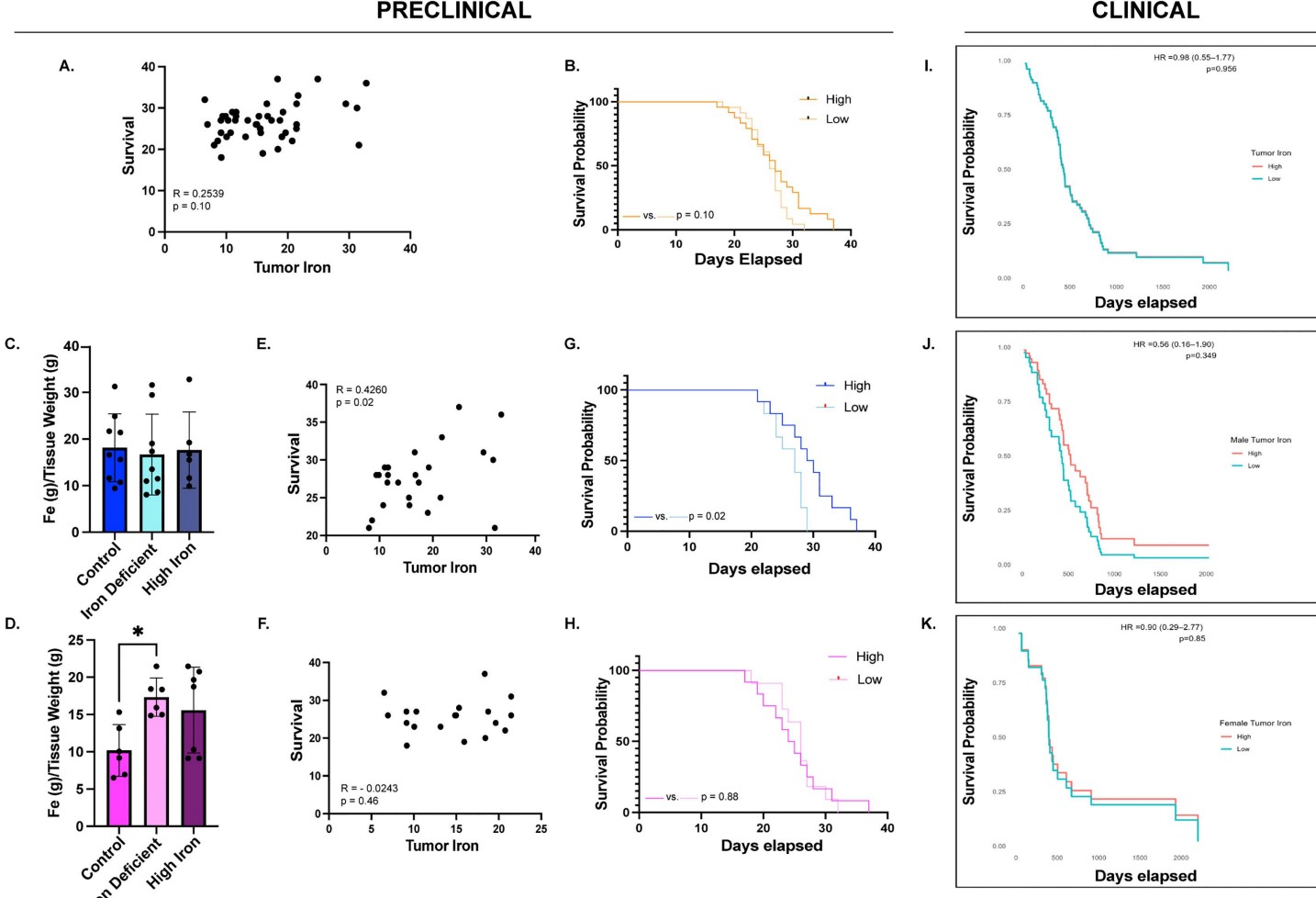

**Fig 5. Tumor iron and survival analyses. A)** Correlations between tumor iron and survival in the combined cohort: No significant correlation between tumor iron and survival was observed (n = 43) (R = 0.2539, p = 0.10). **B)** Kaplan-Meier survival analysis comparing high versus low tumor iron in the combined cohort: High and low tumor iron groups defined using the sex-specific median tumor iron values. No significant survival differences were observed (High iron n = 27; Low iron n = 26) (p = 0.10). **C-D)** Mean tumor iron levels (±SD) in males and females across dietary groups: Tumor iron did not differ among dietary groups in males (Control n = 9; IDA n = 9; High iron n = 6). In females, iron deficiency anemia (IDA) (n = 6) resulted in significantly higher tumor iron compared to controls (n = 6) (*p = 0.02), while high iron diet values did not differ significantly from controls (n = 7) (p = 0.07). Tumor iron data were evaluated for statistical significance using a one-way ANOVA with a Dunnett's multiple comparisons test. **E-F)** Correlation between tumor iron and survival in males and females analyzed separately: Tumor iron significantly correlated with survival in males (n = 24) (R = 0.4260, *p = 0.02), but not in females (n = 19) (R = −0.0243, p = 0.46). **G-H)** Kaplan-Meier survival analysis comparing high versus low tumor iron in males and females separately: Males with high tumor iron exhibited a significant survival advantage (*p = 0.02), whereas no significant difference was observed in females (p = 0.88). **I-K)** Kaplan-Meier survival analysis of patient tumor samples in the combined cohort and in males and females separately: No significant differences in survival advantage were observed in the combined cohort (p = 0.96), or in either males (p = 0.35) or females (p = 0.85).

## Serum iron is altered by diet but not predictive of survival

While it is well established that serum iron is lower under anemic conditions compared to controls [23], the effects of circulating iron under ID and high iron conditions in GBM have not been fully explored. Serum iron levels were measured at tumor endpoint, and there was no significant correlation between serum iron and survival across all mice (R = −0.0528, p = 0.38), nor when analyzed by sex (males: R = 0.2940, p = 0.11; females: R = −0.3004, p = 0.12), although the direction of the correlation differed between sexes (Figs 6B-D). Additionally, both males and females on ID diets tended to have lower serum iron compared to control and high-iron groups, confirming effective iron depletion (Fig 6A). Notably, IDA females had significantly lower serum iron compared to the high iron group (47.97 ± 32.23 vs 95.55 ± 36.62 g/mL, p = 0.04) (Fig 6A). Kaplan-Meier survival analysis comparing mice with high versus low serum iron showed no inherent survival benefit (26.5 vs 27 days, p = 0.60) (Fig 6E). However, when males and females were evaluated separately, there were opposing relationships. Male mice with high serum iron had a slight survival advantage over males with low serum iron whereas high serum iron was slightly disadvantageous in females (25 vs 27 days, p = 0.09) (Figs 6F-G), but these trends were not significant in the preclinical model. Similarly, no evidence of a significant survival association with serum iron was observed in the clinical cohort for the combined cohort (HR = 0.72 (0.36–1.46), p = 0.37) or males or females (males: HR = 0.84 (0.29–2.44), p = 0.75; females: HR = 0.84 (0.21–3.31), p = 0.80) (Figs 6H-J).

## Differential expression of tumor iron uptake and storage proteins between diet groups

To explore potential mechanisms underlying the elevated tumor iron observed in IDA females, we examined expression levels of TfR1, FTH, and FTL. IDA females exhibited a significant increase in TfR1 expression (p = 0.02) that corresponded to the elevated tumor iron (Fig 7A). In males, TfR1 levels were also significantly increased under IDA conditions compared to control (p = 0.02) (Fig 7B). No significant differences were observed between high iron and control diets in females (p = 0.91) or males (p = 0.61). Next, we examined FTL expression as a measure of tumor iron storage via Western blot. No significant differences in FTL expression were observed between ID and control groups in either females (p = 0.67) or males (p = 0.44) (Figs 7C-D). Additionally, there were no significant differences observed between control and high iron groups in either females (p = 0.82) or males (p = 0.64).

Because TfR1 mediates iron uptake primarily through transferrin and FTH, we next examined tumor FTH expression. Consistent with the tumor iron findings, IDA and high iron females had significantly elevated tumor FTH compared to controls (IDA: 104.5 ± 41.54 vs 45.32 ± 33.39 pg/mL, p = 0.02; high iron: 143.5 ± 46.79 vs 45.32 ± 33.39 pg/mL, p = 0.0003), whereas no such differences were observed in males (IDA: 108.1 ± 76.08 vs 133.6 ± 101.5, p = 0.75; high iron: 135.1 ± 56.94 vs 133.6 ± 101.5, p = 0.99) (Figs 7E-F). Despite this, tumor FTH strongly associated with outcome. Higher tumor FTH correlated positively with survival (R = 0.6571, p = 0.0002), and males with high tumor FTH survived significantly longer than those with low levels (29 vs 27 days, p = 0.01) (Figs 7H and K). Corresponding to the finding that tumor iron did not impact survival in females, tumor FTH similarly did not correlate to survival in females (R = 0.08908; p = 0.34), and high versus low tumor FTH did not provide any benefit or detriment (25.5 vs 25 days, p = 0.48) (Figs 7I and L). When examining tumor FTH expression in the clinical tumors, high tumor FTH showed a slight survival advantage in males when compared to low tumor FTH, whereas females had a stronger and opposing trend, though neither were statistically significant (males: HR = 0.52 (0.18–1.53), p = 0.24; females: HR = 2.25 (0.34–15.02), p = 0.40) (Figs 7N-O). No significant trends in survival were observed in the combined cohort (HR = 0.65 (0.29–1.45), p = 0.30) (Fig 7M).

## Distinct associations between serum FTL and FTL with GBM survival outcomes

Next, we evaluated circulating serum FTH and FTL, as measures of iron storage, to assess their relationship with GBM outcomes in this model. Dietary effects within each sex revealed no significant differences in serum FTH (Fig 8A). There was no significant correlation between serum FTH and survival across all mice (R = 0.0099, p = 0.48), nor when analyzed

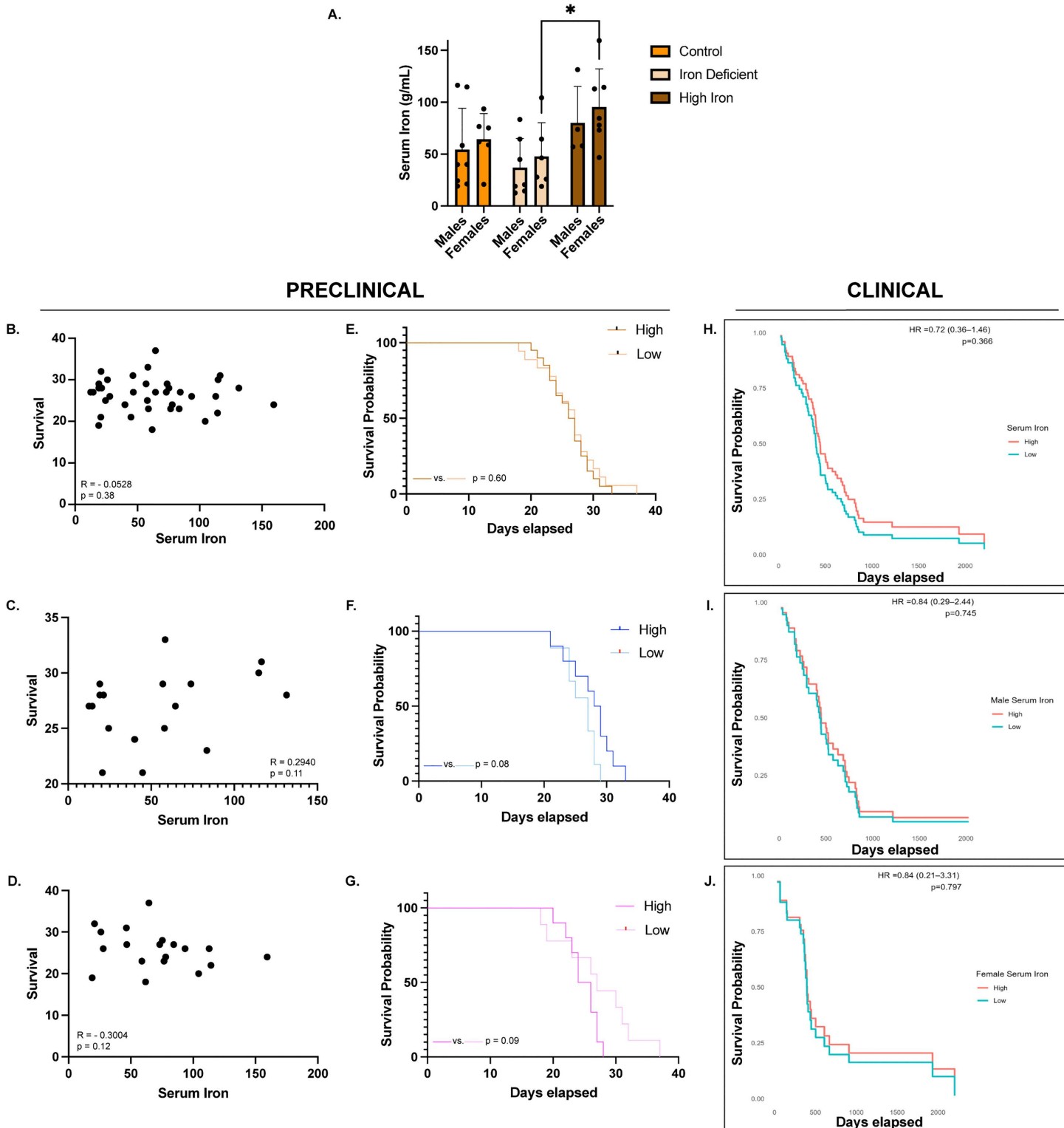

**Fig 6. Serum iron. A)** Mean serum iron levels (±SD) in males and females across dietary groups: Within the females, serum iron differed significantly between the iron deficiency anemia (IDA) (n=6) and high iron diet groups (n=7) (*p=0.04). No significant differences were observed between control (n=6) and either diet in females, or in any male groups (Control n=8; IDA n=7; High iron n=4). Serum iron levels also did not differ between males and females. Statistical significance was assessed using two-way ANOVA with a Tukey's multiple comparison. **B-D)** Correlation between serum iron

and survival in the combined cohort and in males and females separately: Serum iron did not correlate with survival in the combined cohort (n = 38) (R = −0.0528, p = 0.38), in males (n = 19) (R = 0.2940, p = 0.11) or in females (n = 19) (R = −0.3004, p = 0.12). Correlations were evaluated for statistical significance using the Spearman r coefficient of correlation. **E-G)** Kaplan-Meier survival analysis comparing high versus low serum iron in the combined cohort and in males and females separately: High and low serum iron groups were defined using the sex-specific median values. No significant survival differences were observed in any group, although opposing survival trends were noted between males and females. **H-J)** Kaplan-Meier survival analysis of high versus low serum iron in patient samples in the combined cohort and in males and females separately: No significant differences in survival were observed in the combined cohort (p = 0.37), in males (p = 0.75) or in females (p = 0.80).

by sex (males: R = 0.1819, p = 0.23; females: R = −0.02040, p = 0.21) (Figs 8B-D). Kaplan-Meier analysis confirmed no survival differences based on high or low serum FTH in the combined cohort (p = 0.93), in males (p = 0.42) or females (p = 0.62) individually (Figs 8E-G). Similarly, no significant differences were observed in the clinical cohort when analyzed together (HR = 0.76 (0.36–1.61), p = 0.48) or separately (males: HR = 0.68 (0.22–2.07), p = 0.50; females: HR = 2.35 (0.19–29.00) p = 0.51) (Figs 8H-J). While not statistically significant, a trend towards prolonged survival emerged in females with lower serum FTH at later timepoints.

In contrast, serum FTL showed significant diet-dependent differences. IDA males had lower serum FTL than controls (125,400 ± 71,328 vs 354,938 ± 125,420 pg/mL, p = 0.03) and high iron mice (125,400 ± 71,328 vs 509,875 ± 110,805 pg/mL, p = 0.0008) (Fig 9A). IDA females also had lower serum FTL than controls (141,833 ± 67,682 vs 442,700 ± 121,078 pg/mL, p = 0.004) and high iron females (141,833 ± 67,682 vs 470,500 ± 174,894 pg/mL, p = 0.0005) (Fig 9A). No differences were observed between males and females within the same diet, or between control and high iron groups. Correlation analysis revealed no significant association between serum FTL and survival when considering all mice (R = 0.0395, p = 0.41) or females alone (R = −0.2233, p = 0.19) (Figs 9B and D). However, high serum FTL in males significantly correlated with increased survival (R = 0.6889, p = 0.002) (Fig 9C). Consistent with these findings, Kaplan-Meier analysis revealed sex-specific survival differences in our preclinical model, where high serum FTL conferred a significant survival advantage in males (29 vs 24.5 days, p = 0.003) and a significant disadvantage in females (23 vs 27 days, p = 0.04) (Figs 9F-G). No differences were observed in the combined cohort based on serum FTL levels (26.5 vs 26 days, p = 0.82) (Fig 9E). In the clinical samples, serum FTL did not provide a survival advantage in either direction for males (HR = 0.78 (0.22–2.70), p = 0.69). However, low serum FTL, similar to the preclinical model, provided a survival benefit for females, though not statistically significant (HR = 15.75 (0.46–538.20), p = 0.13) (Figs 9H-J).

## Discussion

In this study, we investigated the impact of dietary manipulation of systemic iron status on tumor iron handling and survival in GBM, integrating findings from both an immune-competent mouse model and clinical patient samples.

Although the clinical analyses did not yield statistically significant associations between serum or tumor iron/ferritin and survival, these exploratory experiments paralleled several key findings in the mouse model, providing important translational context. One explanation for this is that the clinical samples were collected at earlier stages of disease at the time of initial resection, whereas preclinical samples were obtained at endpoint, allowing the preclinical model to capture changes in tumor iron at different stages of disease. Additionally, while we did control for treatment in clinical analyses, it is possible that the different treatment types may impact tumor and serum iron dynamics. Despite these differences, the preclinical model demonstrated clear sex-dependent effects, with the clinical data showing parallel, but not statistically significant, directionality, highlighting the benefit of integrating both models to generate hypotheses regarding iron regulation in GBM.

In the preclinical model, IDA males had reduced survival compared to controls complimented by increased tumor volumes, while IDA females showed significantly smaller tumors and a modest survival advantage. Notably, earlier tumor volume measurement (10 days post injection) in this model provided insight into tumor establishment and iron-dependent growth dynamics. At this stage, several male control and female IDA mice did not yet exhibit measurable tumors, suggesting delayed tumor initiation or early growth restraint in these groups that may have impacted downstream survival

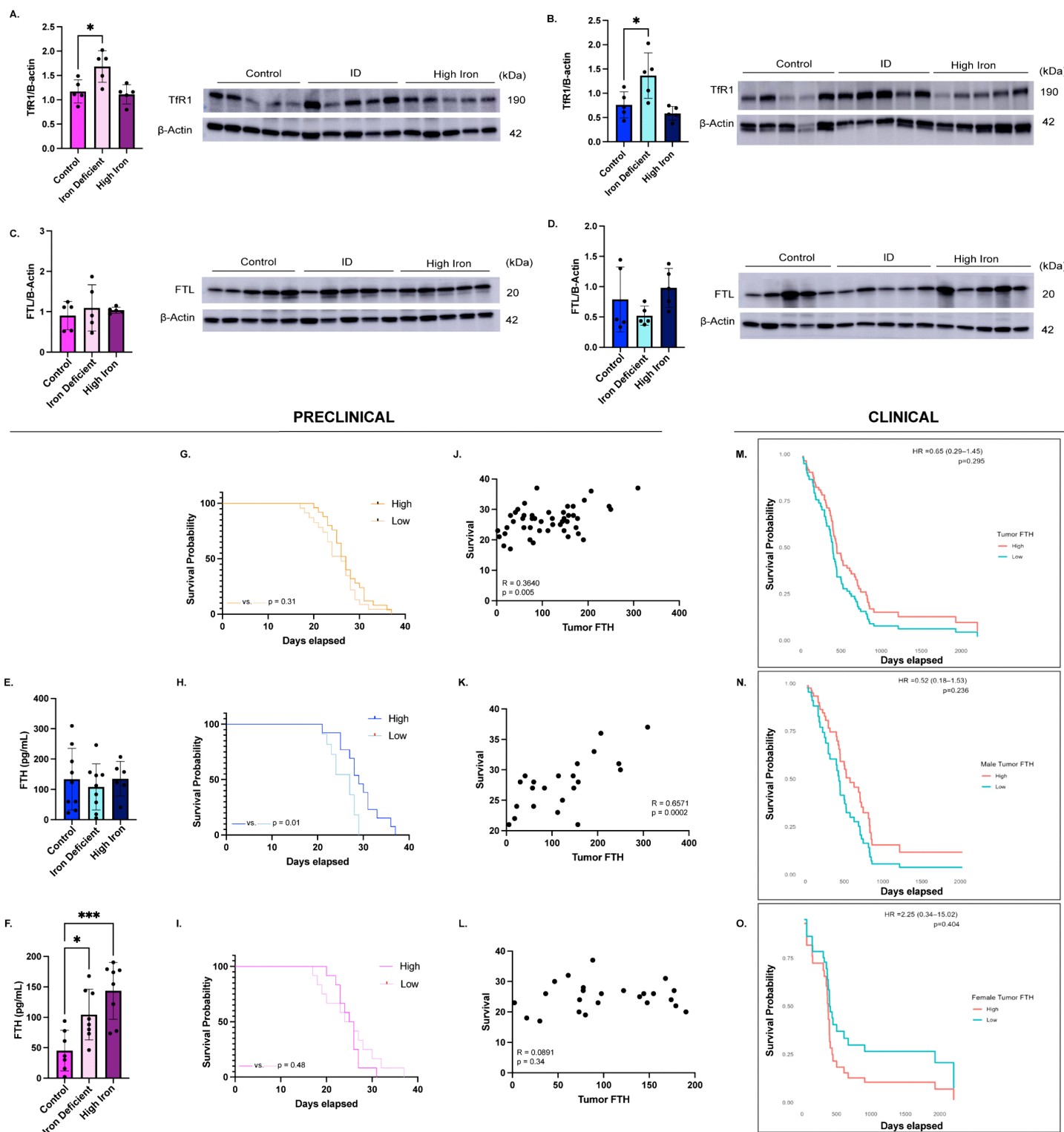

**PRECLINICAL**

**CLINICAL**

**Fig 7. Tumor iron-related protein expression and survival analyses. (A-B)** Transferrin receptor (TfR1) expression in male and female tumors with representative immunoblots: Iron deficiency anemia (IDA) resulted in significantly increased TfR1 expression compared to controls in both males (*p = 0.02) and females (*p = 0.02). N = 5 mice per group, normalized to β-actin. Statistical significance was assessed using one-way ANOVA with a Dunnett's multiple comparisons test. **C-D)** Ferritin light chain (FTL) expression in male and female tumors with representative immunoblots: FTL expression

did not differ among dietary groups in either sex. N = 5 mice per group, normalized to β-actin. Statistical significance was assessed using one-way ANOVA with a Dunnett's multiple comparisons test. **E-F)** Ferritin heavy chain (FTH) protein concentration in male and female tumors measured by enzyme-linked immunosorbent assay (ELISA): In females, IDA (n = 8) (*p = 0.02) and high iron (n = 8) (***p = 0.0003) resulted in a significant increase in tumor FTH compared to controls (n = 7). No significant differences were observed in IDA males (n = 9) (p = 0.75) or high iron males (n = 6) (p = 0.99) when compared to controls (n = 9). Statistical significance was assessed using one-way ANOVA with a Dunnett's multiple comparisons test **G-I)** Kaplan-Meier survival analysis comparing high versus low tumor FTH in the combined cohort and in males and females separately: High and low groups defined using the sex-specific median tumor FTH values. Males with high tumor FTH had a significant survival advantage over those with low levels (*p = 0.01). No significant differences were observed in the combined cohort (p = 0.31) or in females (p = 0.48). **J-L)** Correlation between tumor FTH and survival in the combined cohort and in males and females analyzed separately: Tumor FTH significantly correlated to survival in the combined group (n = 48) (R = 0.3640, **p = 0.005) and in males (n = 24) (R = 0.6571, ***p = 0.0002), but not in females (n = 24) (R = 00891, p = 0.34). Correlations were assessed for statistical significance using the Spearman r coefficient of correlation. **M-O)** Kaplan-Meier survival analysis of patient tumor samples in the combined cohort and in males and females separately: No significant survival differences were observed in the combined cohort (p = 0.30), males (p = 0.24), or females (p = 0.40).

outcomes. In contrast to previous clinical studies evaluating iron supplementation, increasing dietary iron in our model did not significantly alter tumor growth or survival in either sex [13]. Under ID conditions, TfR1 expression is increased to enhance iron uptake and maintain iron homeostasis, positioning the body to rapidly absorb iron upon repletion. This mechanism has been shown to vary between sexes in the brain but may help explain the clinical findings of improved survival following iron supplementation in anemic patients [24]. Our results, in contrast, showed no survival advantage from placing control mice on high iron diets, suggesting that high iron, either through diet or supplementation, may only provide a survival benefit in the case of a pre-existing deficiency.

Contralateral hemispheric brain iron further supported a sex-dependent regulation in systemic versus local brain iron in this animal model. Consistent with prior literature, IDA males had significantly lower iron content compared to high iron and control diets in the contralateral hemisphere, whereas the females remained stable, independent of systemic iron status [24]. Brain iron status is dependent on distinct signaling at the blood-brain barrier (BBB), and this signaling may change throughout the course of GBM progression [25–27]. Additionally, this regulation is further influenced by tumor cell state, with non-stem like cells increasing BBB permeability to a greater extent than stem-like cells [28]. Although BBB function was not assessed here, the sex-specific patterns in brain iron are consistent with differences in BBB iron handling, suggesting that systemic iron alone cannot predict brain iron content.

Clinical tumor iron measurements also revealed subtle but informative survival trends. Males with higher tumor iron survived slightly longer than those with low tumor iron, whereas this relationship was opposite amongst females, though it was minimal. Similar directionality was observed for tumor FTH, where high FTH may provide a slight benefit for males, with the opposite trend occurring in females. These observations are much more pronounced in the preclinical mouse model, but the relationships remain aligned between groups, with high tumor iron and FTH providing a significant survival benefit in males, but not females. These patterns suggest a regulation of tumor iron acquisition that may be consistent with the clinical data, though the mechanisms and timing of these effects likely differ throughout disease progression and treatment course.

When examining tumor iron directly between dietary groups in the preclinical model, our analysis revealed several alterations in iron handling between diets, with a distinct sex effect. In females, IDA significantly increased tumor iron levels compared to control. Females on high iron diets had slightly higher tumor iron, but this result was not significant and likely reflects increased circulating serum iron. Given that TfR1 is the primary mediator of cellular iron uptake, either through transferrin-bound iron and/or ferritin-bound iron, we sought to determine if these differences were linked to changes in TfR1 expression [29]. Our analyses revealed a significant increase in TfR1 expression in the tumors of IDA males compared to controls, without a corresponding increase in FTH. This elevation in TfR1 is consistent with a compensatory response to iron deficiency, where TfR1 is upregulated to enhance iron uptake and restore homeostasis. However, given the reduced iron availability systemically and in the contralateral hemisphere of IDA males, the availability of iron to support this response is likely limited. In contrast, both TfR1 and FTH were significantly increased in IDA female tumors,

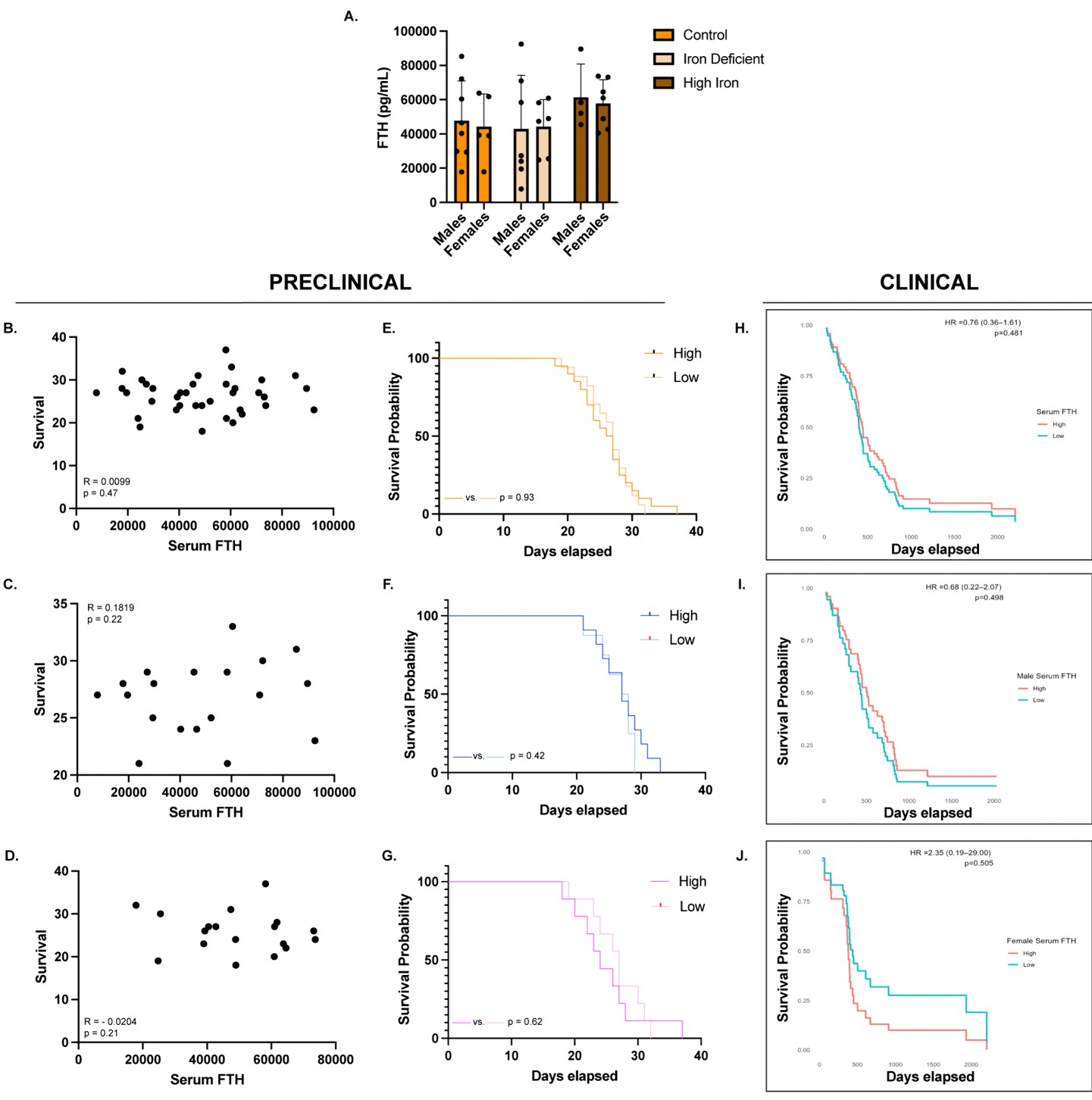

**Fig 8. Serum ferritin heavy chain A) Ferritin heavy chain (FTH) protein concentration in male and female serum measured by enzyme-linked immunosorbent assay (ELISA): No significant differences were observed amongst any of the analyzed groups or between sex (Males: Control n = 8; IDA n = 7; High iron n = 4; Females: Control n = 5; IDA n = 6; High iron n = 7).** Serum FTH levels were evaluated for statistical significance using two-way ANOVA with a Tukey's multiple comparisons test. **B-D)** Correlation between serum FTH and survival in the combined cohort and in males and females separately: Serum FTH did not significantly correlate to survival in the combined cohort (n = 37) (R = 0.0099, p = 0.48), in males (n = 19) (R = 0.1819, p = 0.23) or in females (n = 18) (R = −0.0204, p = 0.21). Correlations were evaluated for statistical significance using the Spearman r coefficient of correlation. **E-G)** Kaplan-Meier survival analysis comparing high versus low serum FTH in the combined cohort and in males and females separately:

High and low serum FTH groups were defined using the sex-specific median values. No significant survival differences were observed in any group. **H-J)** Kaplan-Meier survival analysis of high versus low serum FTH in patient samples in the combined cohort and in males and females separately: No significant differences in survival were observed in the combined cohort (p = 0.48), in males (p = 0.50) or in females (p = 0.51).

consistent with enhanced iron uptake and storage. Given that prior studies report no changes in FTH or TfR1 expression in IDA female brains, these alterations are likely a tumor-specific mechanism that reflects dietary iron rather than contralateral iron availability [24]. The underlying mechanism, however, remains unclear, as the tumor iron content and FTH expression observed in this study suggests increased iron availability, yet the elevation in TfR1 expression would suggest the opposite. One possible explanation is post-transcriptional regulation of TfR1, as several microRNAs, including miR-148, are known to modulate TfR1 expression and are downregulated in GBM tissue and cell lines [30,31]. Additionally, GBM tumors are very heterogenous, typically comprised of tumor cells, immune cells, and resident brain cells. Even in our preclinical mouse model that utilizes only a single cell type to initiate tumor growth, diverse cells populations still accumulate during progression [32]. Therefore, while the changes in protein expression observed here support altered tumor iron handling in IDA females, discerning the mechanisms, including enhanced tumor iron import, specific metabolic adaptations, or redistribution between these diverse cell types, will require future work to determine cell-specific resolution.

In our current analysis of clinical GBM samples, no statistically significant associations were observed between high and low iron or ferritin levels within the serum, which contrasts previous reports that have identified a robust relationship between serum iron and survival in a smaller patient cohort [33]. Many studies examining the role of anemia in GBM produce variable results, highlighting the complexity of interpreting clinical measures in heterogenous patient populations [34,35]. Notably, our analysis controlled for the use of dexamethasone, a clinical factor known to influence white blood cell count and patient outcomes in GBM [36,37]. This additional factor may partially explain the observed discrepancy. Importantly, however, many of the underlying trends from the animal models in this highly controlled setting held true. For example, in both cases, males with high tumor iron survived slightly longer than those with low tumor iron, but this was not true in females. Serum ferritin also showed some similarities between models: In humans, FTH and FTL conferred very little survival advantage in males. In females, there was a stronger but nonsignificant survival advantage with low serum FTH and FTL, but no observed survival trend for serum iron levels. These patterns reflect the preclinical findings, where high serum FTL, but not FTH, was significantly associated with improved survival in males, and low serum FTL was significantly associated with improved survival in females.

In the preclinical model, we compared serum iron levels between control, IDA, and high iron mice. While the animals were anemic based on hemoglobin and hematocrit measures, serum iron did not differ significantly between control and iron deficient groups or control and high iron groups. It should be noted, however, that high iron males had a 68% increase in serum iron compared to controls, while females exhibited a 67% increase. These increases were not complimented by changes in survival or iron levels in the tumor or brain tissue, further suggesting that systemic iron availability does not directly translate to tissue-specific iron levels. Similarly, serum FTH levels did not significantly differ between control and IDA groups. Although both ferritin subunits participate in iron storage, they differ in regulation and physiological role in circulation. FTL is considered the primary indicator of iron storage, while FTH is more strongly regulated by oxidative stress and inflammatory signals [38,39]. The reduction in serum FTL observed in our preclinical model therefore suggests that IDA alters iron storage rather than just circulating iron. This depletion may also limit iron availability to peripheral tissues, including the tumor microenvironment, despite preserved serum iron. Overall, our results point to a previously unrecognized sex-dependent link between systemic and tumor iron across both clinical and preclinical data, strengthening the translational relevance of our model and highlighting the need to further investigate temporal shifts in iron dynamics during GBM progression.

A key strength of this study is the use of a translatable, immune-competent model that recapitulates several clinical features of GBM alongside human GBM clinical samples. Notably, several observations in our preclinical model are

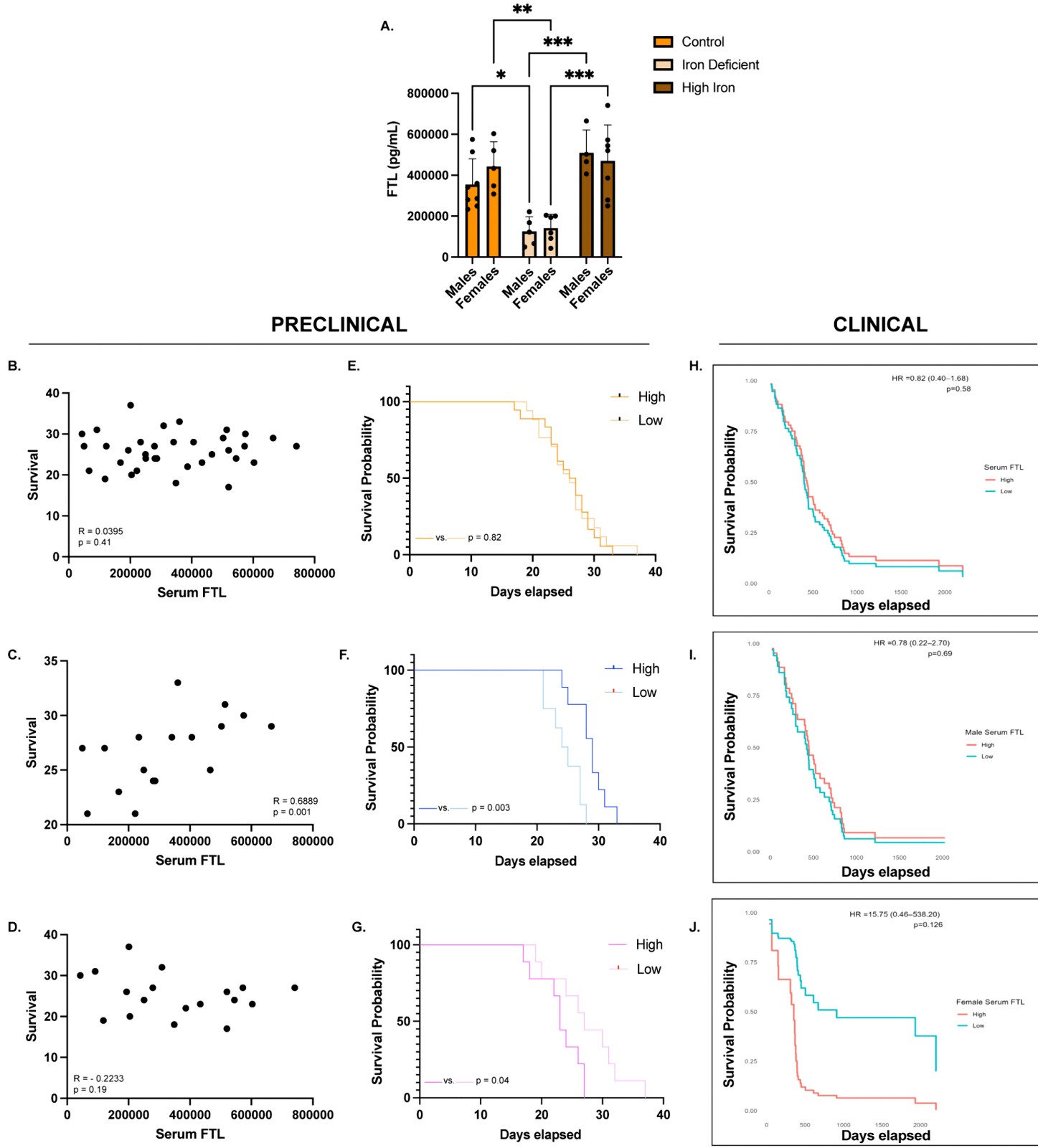

**Fig 9. Serum ferritin light chain A)** Ferritin light chain (FTL) protein concentration in male and female serum measured by enzyme-linked immunosorbent assay (ELISA): Iron deficient (ID) males (n=5) had significantly reduced serum FTL compared to control (n=8) (*p=0.03) and

**high iron (n = 4) (\*\*\*p = 0.0008) groups. ID females (n = 6) also had significantly reduced serum FTL compared to control (n = 5) (\*\*p = 0.004) and high iron (n = 7) (\*\*\*p = 0.0005) groups.** Serum FTL levels were evaluated for statistical significance using two-way ANOVA with a Tukey's multiple comparisons test. **B-D)** Correlation between serum FTL and survival in the combined cohort and in males and females separately: Serum FTL did not significantly correlate to survival in the combined cohort (n = 35) (R = 0.0395, p = 0.41) or in females (n = 18) (R = −0.2233, p = 0.19). Serum FTL significantly correlated to survival in males (n = 17) (R = 0.6889, p = \*\*\*0.001). Correlations were evaluated for statistical significance using the Spearman r coefficient of correlation. **E-G)** Kaplan-Meier survival analysis comparing high versus low serum FTL in the combined cohort and in males and females separately: High and low serum FTL groups were defined using the sex-specific median values. Males with high serum FTL had a significant survival advantage compared to low serum FTL (\*\*p = 0.003). Females with high serum FTL had a significant survival disadvantage compared to low serum FTL (\*p = 0.04). No significant differences in survival were observed in the combined cohort (p = 0.82). **H-J)** Kaplan-Meier survival analysis of high versus low serum FTL in patient samples in the combined cohort and in males and females separately: No significant differences in survival were observed in the combined cohort (p = 0.48), in males (p = 0.50) or in females (p = 0.51), though a strong relationship was observed in females that aligned with the preclinical results.

consistent with previously described literature, including the direction of the survival trends in IDA patients, reflecting reported sex differences [13,14]. This study design allowed us to manipulate circulating iron levels using three diets of varying iron concentrations to determine the influence of systemic iron on tumor iron biology. By including both sexes and examining multiple iron handling proteins in both serum and tumor, our data helps to clarify previous ambiguity in iron dynamics in GBM. The preclinical model of iron deficiency employed here produces mild to moderate anemia due to the general limitations of dietary models at this age of animals [40]. Interestingly, we still saw significant changes in many of the parameters that were measured despite this, suggesting that even minor changes to circulating iron can modulate tumor growth. Additionally, anemia in cancer patients is typically mild, affecting about 90% of individuals, which aligns with the observed phenotype of our model [41]. In the clinical samples, only ~14% of patients (4 females and 6 males) were anemic, and thus we could not perform some of the more direct analyses seen with the preclinical model due to sample size. Hemoglobin levels below 10 g/dL, indicating moderate to severe anemia, are more commonly observed (~40%) as the disease progresses, whereas these samples were taken early, likely explaining the low number of anemic patients. Some studies point to the onset of anemia as a result of increased inflammation and shortened red blood cell survival, while others suggest chemotherapy is the main driver through destruction of erythrocytes [42,43]. We measured hemoglobin and hematocrit at the endpoint in our untreated mice and saw data supporting the latter, as hemoglobin measures in IDA mice either remained at their pre-tumor levels or were slightly increased between initial and final measurements in the absence of chemotherapy (S1 Fig A-D). Future work incorporating the standard-of-care treatments will be necessary to determine whether systemic iron levels influence therapeutic response or vice versa.

In conclusion, this study indicates that systemic iron levels, through IDA or high iron plays a significant role in modulating tumor dynamics in GBM. Previous literature has established several links between iron and GBM outcomes, though several gaps remained in understanding systemic and sex-specific effects. Here, we addressed these gaps by linking iron levels and related proteins to survival outcomes in human and murine models, revealing key sex-dependent differences. Although the human correlations were not statistically significant in this exploratory study, the consistency of these directional trends support the translational value of the mouse model and implies that the relationships may strengthen as the disease progresses. Thus, while human samples captured early, less aggressive disease, the preclinical model provided information on advanced stage iron dynamics, allowing the data to inform one another across the disease spectrum.

## Supporting information

**S1 Fig. Hemoglobin and Hematocrit Levels at experimental endpoint. A-B)** Mean hemoglobin and hematocrit measures in males (±SD): At the experimental endpoint, there were no significant differences in hemoglobin between control and iron deficiency anemia (IDA) mice (p = 0.83) or high iron mice (p = 0.18). There were also no significant differences in hematocrit between control and IDA mice (p = 0.80) or high iron mice (p = 0.12). **C-D)** Mean hemoglobin and hematocrit measures in females (±SD): At the experimental endpoint, IDA females exhibited significantly reduced hemoglobin

(**p = 0.006) and hematocrit (**p = 0.002) compared to controls. There were no significant differences in the high iron group compared to controls for hemoglobin (p > 0.99) or hematocrit (p = 0.79).
(TIF)

**S2 Fig. Kaplan-Meier survival curves prior to multivariate cox proportional hazards regression in the combined cohort. A)** Kaplan-Meier survival analysis of clinical samples of high versus low serum iron. No significant differences in survival were observed (p = 0.75) **B)** Kaplan-Meier survival analysis of high versus low serum ferritin heavy chain (FTH). No significant differences in survival were observed (p = 0.42). **C)** Kaplan-Meier survival analysis of high versus low serum ferritin light chain (FTL). No significant differences in survival were observed (p = 0.68). **D)** Kaplan-Meier survival analysis of high versus low tumor iron. No significant differences in survival were observed (p = 0.57) **E)** Kaplan-Meier survival analysis of high versus low tumor FTH. No significant differences in survival were observed (p = 0.53).
(TIF)

**S3 Fig. Kaplan-Meier survival curves prior to multivariate cox proportional hazards regression in males. A)** Kaplan-Meier survival analysis of male clinical samples of high versus low serum iron. No significant differences in survival were observed (p = 0.75). **B)** Kaplan-Meier survival analysis of high versus low serum ferritin heavy chain (FTH). No significant differences in survival were observed (p = 0.96). **C)** Kaplan-Meier survival analysis of high versus low serum ferritin light chain (FTL). No significant differences in survival were observed (p = 0.46). **D)** Kaplan-Meier survival analysis of high versus low tumor iron. No significant differences in survival were observed (p = 0.31) **E)** Kaplan-Meier survival analysis of high versus low tumor FTH. No significant differences in survival were observed (p = 0.51).
(TIF)

**S4 Fig. Kaplan-Meier survival curves prior to multivariate cox proportional hazards regression in females. A)** Kaplan-Meier survival analysis of female clinical samples of high versus low serum iron. No significant differences in survival were observed (p = 0.87). **B)** Kaplan-Meier survival analysis of high versus low serum ferritin heavy chain (FTH). No significant differences in survival were observed (p = 0.32). **C)** Kaplan-Meier survival analysis of high versus low serum ferritin light chain (FTL). No significant differences in survival were observed (p = 0.30). **D)** Kaplan-Meier survival analysis of high versus low tumor iron. No significant differences in survival were observed (p = 0.79) **E)** Kaplan-Meier survival analysis of high versus low tumor FTH. No significant differences in survival were observed (p = 0.98).
(TIF)

**S5 Fig. Raw Western Blot Images for Fig 7A-D.** Full uncropped images of transferrin receptor (TfR1, approx. 190kD), ferritin light chain (FTL, approx. 20kD), and corresponding β-actin (approx. 42kD) in tumor lysates in males and females. Lanes 1–5: Control; Lanes 6–10: Iron Deficient; Lanes 7–15: High Iron. Loading schemes were kept consistent for all gels.
(PDF)

**S1 Table. Statistical analysis for preclinical data.** Statistical analysis for all figures was performed using GraphPad Prism. Source data and statistical outputs for each figure are provided in the accompanying Excel file, with individual tabs labeled by the corresponding figure panel. This includes the statistical test used, exact p-values, groupwise n, and additional descriptive statistics.
(XLSX)

**S2 Table. Statistical analysis for clinical data.** Multivariate cox proportional hazard regression of clinical outcomes for all factors. Hazard ratios, 95% confidence intervals, and associated p-values are reported for all factors. The regression was performed controlling for relevant clinical covariates such as patient sex, age at diagnosis, white blood cell count, extent of tumor resection, receipt of chemotherapy, dexamethasone, and/or radiation.
(DOCX)

## Acknowledgments

The authors thank Tammy Hyde and Ellen Mullady in the Department of Comparative Medicine at the Penn State College of Medicine for their contributions and expertise in hematology. Xiaoyu Wang at the MRI Core for their expertise in image acquisition. The MRI Core (RRID:SCR_021198) services and instruments used in this project were funded, in part by the Pennsylvania State University College of Medicine via the Office of the Vice Dean of Research and Graduate Students and the Pennsylvania Department of Health using Tobacco Settlement Funds (CURE). The content is solely the responsibility of the authors and does not necessarily represent the official views of the university or College of Medicine. The Pennsylvania Department of Health specifically disclaims responsibility for any analyses, interpretations, or conclusions. The authors also thank Laura Liermann, Michael Lawrence Robbins, and Maggie Wang from the Penn State University Laboratory for Isotopes and Metals in the Environment for their analyses using ICP-AES and ICP-MS. Elizabeth Neely and Becky Webb for technical assistance.

## Author contributions

**Conceptualization:** Emily Tufano, James R. Connor.

**Data curation:** Emily Tufano.

**Formal analysis:** Emily Tufano, Kondaiah Palsa, Rebecka O. Serpa, Timothy B. Helmuth, Aurosman Sahu.

**Funding acquisition:** James R. Connor.

**Investigation:** Emily Tufano, Kondaiah Palsa, Rebecka O. Serpa, Timothy B. Helmuth, Sara Mills-Huffnagle, Mathias Kant, Aurosman Sahu.

**Methodology:** Emily Tufano, Kondaiah Palsa, James R. Connor.

**Project administration:** Emily Tufano.

**Resources:** Gabriela Remit-Berthet, James R. Connor.

**Validation:** Emily Tufano.

**Visualization:** Emily Tufano.

**Writing – original draft:** Emily Tufano, James R. Connor.

**Writing – review & editing:** Emily Tufano, Kondaiah Palsa, Rebecka O. Serpa, Timothy B. Helmuth, Gabriela Remit-Berthet, Sara Mills-Huffnagle, Mathias Kant, Aurosman Sahu, James R. Connor.

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
