## [Decision Letter · Decision Letter 0]

19 Jan 2026

Dear Dr. Tufano,

Thank you for submitting your manuscript to PLOS ONE. After careful consideration, we feel that it has merit but does not fully meet PLOS ONE’s publication criteria as it currently stands. Therefore, we invite you to submit a revised version of the manuscript that addresses the points raised during the review process.

We look forward to receiving your revised manuscript.

Kind regards,

Kostas Pantopoulos, PhD

Academic Editor

PLOS One

**Journal Requirements:**

4. We are unable to open your Supporting Information file [File Name]. Please kindly revise as necessary and re-upload.

Reviewers' comments:

Reviewer's Responses to Questions

**Comments to the Author**

1. Is the manuscript technically sound, and do the data support the conclusions?

Reviewer #1: Partly

Reviewer #2: Partly

2. Has the statistical analysis been performed appropriately and rigorously?

Reviewer #1: I Don't Know

Reviewer #2: No

3. Have the authors made all data underlying the findings in their manuscript fully available?

Reviewer #1: No

Reviewer #2: Yes

4. Is the manuscript presented in an intelligible fashion and written in standard English?

Reviewer #1: Yes

Reviewer #2: Yes

Reviewer #1: The manuscript by Tufano et al describes a study analysing the role of iron in glioblastoma, using a mouse model and a cohort of patient samples. It is a careful study with complex, and some robust findings in the animal model, but clinical data is overinterpreted.

Below are detailed comments and questions on the manuscript.

Is there an overlap of clinical GBM samples/data with previous study (Shenoy Transl Oncol 2025)?

Do mice have similar sex differences as humans with regard to iron levels and metabolism? Is this a relevant model?

Overall, the order of figures is confusing and could be made more logical.

Human ethics Define ‘honest broker’

Animal ethics Acceptable

Title: Should the title reflect the importance of sex differences?

Line 39 Overinterpretation – there is no evidence from clinical data for any association with patient survival

Line 56 Oxygen production?

Line 112 Method clarification needed: What diet was the high-iron group on prior to tumour implant? Were the control and IDA groups maintained on their respective diets (since PND 21) post tumour implant?

Line 175 A loss of 9/50 animals due to monitoring timing seems a waste.

Line 193 Western blot data for FTH1 not shown, why listed here?

Line 228, 246 No data on multivariate cox proportional hazards regression is presented. And sentence repeated.

Fig 3 Why was day 10 chosen? The majority of male mice on control diet did not have measurable tumours. And the majority of female mice on iron deficiency diet similarly showed no tumour. Legend should specify 10 days. No significant difference in survival.

Fig 4 and 5 There is no evidence from mice that high iron diet resulted in different tumour or brain tissue levels compared to controls. The period of dosing may have been too short. Figure 7 shows blood data – suggest moving closer.

Line 346 A rather optimistic heading as only male mice showed this association; female mice and human patients did not.

Line 363 There is no trend in clinical data (p=0.31, 0.57 and 0.79).

Would it be worth investigating the amount of iron the tumour was able to accumulate in relation to (surrounding/contralateral) normal brain?

Tumour contains notably more iron than normal brain. Discuss possible reasons.

Different parts of the brain accumulate different levels of iron; was this considered?

Fig 6 The writing on several graphs is too small to decipher (ie p-values).

Fig 7 No clear difference between control and high iron diet measures.

Line 457 There is no trend in clinical data (p=0.75, 0.58, 0.87).

Clarification: does FTL refer to tissue levels and L-ferritin to serum levels? Similarly, for FTH and H-ferritin?

Fig 8 Ensure graph orientation (x-y) to be the same for B, C and D. p-values for F and G moved off the graph.

Fig 9 Ensure graph orientation to be the same for B, C and D.

Line 503 No evidence that low serum FTL provided advantage to females.

Fig S1 not mentioned in Results.

Line 551 Not convinced that ‘both models demonstrated sex-dependent patterns’

Line 566 ‘consistent with prior literature’ but no references given

Their previous paper (Shenoy Transl Oncol 2025) showed a robust relationship between serum iron and survival outcome in patients with GBM, which was not supported in this bigger patient cohort. Discuss

Conflict of interest: a previous publication noted that ‘J.R.C is a founder and chairman of the board of Siderobioscience LLC a company founded on patented technology for management of iron deficiency’. Should this be noted?

Reviewer #2: This study presents a valuable contribution to the understanding of iron metabolism in GBM, particularly in highlighting sex-specific responses to systemic iron availability. This manuscript combines an immune-competent GL261 intracranial mouse model with analysis of human GBM samples to test whether dietary systemic iron affects tumor/brain iron handling and survival in a sex-dependent manner. he preclinical work is robust and well-executed, and the integration of human data adds translational context. However, the clinical findings should be interpreted as preliminary due to limited sample size and lack of statistical significance. Some of my concerns are indicated below:

1. In preclinical results, many comparisons are based on small group sizes, and several key findings hover near conventional significance thresholds. This raises a risk of type-II and type-I errors and of over-interpreting trends. Please report group-wise N for every panel and include confidence intervals and effect sizes, and consider a pre-specified multiplicity correction (or explicitly justify the testing strategy).

2. Authors state that mice were “arbitrarily placed” on diets at PND21, which is not the same as randomized allocation; IACUC approval is stated, but the manuscript should explicitly describe randomization and whether experimenters were blinded to diet when assessing for example MRI/survival/analyses. Add explicit statements on randomization, allocation concealment, and blinding.

3. In survival analyses, please include multivariable Cox models with interaction terms (sex × iron status, sex × tumor iron) and present HRs with CIs; the Methods mention multivariate Cox models but results are sparse, expand the tables/figures and provide model diagnostic.

4. For ICP measurements and ELISAs provide detection limits, intra/inter-assay CVs, and whether values below LOD occurred; ensure units are consistently reported because some snippets show g/tissue weight and g/mL.

5. The finding that iron-deficient females exhibit increased tumor iron content together with elevated TfR1 and FTH expression is biologically compelling; however, the underlying mechanism remains unclear, as the authors also acknowledge. In its current form, the study does not allow discrimination between alternative, non-mutually exclusive explanations, such as enhanced tumor iron import, tumor cell–intrinsic metabolic adaptations, or redistribution of iron within the tumor microenvironment (for example, accumulation in immune cells with high ferritin content). To strengthen this aspect, the authors should either include additional experiments or more explicitly frame these interpretations as hypotheses for future work. Potential approaches that would help resolve this issue include: (i) iron histochemistry (e.g., Perls’ staining) combined with ferritin immunohistochemistry and cell-type–specific markers, (ii) assessment of systemic iron regulators such as transferrin saturation and hepcidin, (iii) cell-type–resolved iron quantification (e.g., flow cytometry, laser-capture microdissection, or spatial transcriptomics), and/or (iv) evaluation of blood–brain barrier permeability.

.

Reviewer #1: **Yes:** Gabi DachsGabi DachsGabi DachsGabi Dachs

Reviewer #2: No

---

## [Author Response · Author response to Decision Letter 1]

5 Mar 2026

REFEREE 1:

The manuscript by Tufano et al describes a study analysing the role of iron in glioblastoma, using a mouse model and a cohort of patient samples. It is a careful study with complex, and some robust findings in the animal model, but clinical data is overinterpreted.

Below are detailed comments and questions on the manuscript.

1. Is there an overlap of clinical GBM samples/data with previous study (Shenoy Transl Oncol 2025)?

A total of 40 samples overlapped between this study and Shenoy et al. (2025). All processing and analyses in the present manuscript were performed by the investigators on this manuscript using separate aliquots of serum and tumor tissue, following the methods described in Shenoy et al. (2025). Notably, the Shenoy et al. (2025) paper reports including only 21 of these overlapping samples into their survival analysis, but the specific subset is not specified. We report using 52 samples for our corresponding serum analysis.

We discuss discrepancies in these results in response to reviewer’s comment 29 and in the discussion of the manuscript (please see comment #29)

2. Do mice have similar sex differences as humans with regard to iron levels and metabolism? Is this a relevant model?

The reviewer raises a good point on the relevance of the model, and we acknowledge the limitations of our approach. However, there is considerable and substantial information on the relevance of mouse models in iron biology in a recent review entitled: “Regulation of iron homeostasis: Lessons from mouse models,” (PMID: 32792212) though sex differences are not mentioned.

We have added a summary sentence on the relevance of our model in the introduction. In this section and its associated references, are several key points regarding model relevance: In human GBM, males exhibit significantly higher binding of radiolabeled FTH compared to females under baseline conditions. Consistent with this, we observe higher tumor FTH levels in male mice compared with females in the control diet groups, though uptake was not directly measured. Additionally, our preclinical survival findings parallel those reported in Shenoy et al. (2024), including sex-dependent relationships between anemia and survival. In terms of general iron biology, as the reviewer points out in #17 and #18, there is an elevation of tumor iron compared to contralateral brain tissue in our preclinical model. This finding has been cited in the literature for human GBM samples, reporting elevated iron uptake in GBM tissues compared to surrounding brain tissue, likely due to the elevated transferrin receptor.

The revised manuscript now reads:

Introduction: This immune competent model recapitulates key features of GBM, and in our study this model was able to reproduce sex-dependent iron and ferritin patterns observed in patients, supporting its translational relevance [15-17]. To complement these studies, we analyzed human GBM tumor and serum samples for circulating iron and ferritin levels.

3. Overall, the order of figures is confusing and could be made more logical.

In response to the reviewer, we have updated the order of the figures and results in a way that we view as more logical. The story is now described from the iron perspective (brain iron (Figure 4), tumor iron (Figure 5), and serum iron (Figure 6), followed by their related iron proteins (tumor iron proteins (Figure 7) and serum iron proteins (Figure 8 and 9).

Additionally, we have changed the order of the figures within Figure 7 to be consistent with Figures 5-6 and 8-9.

4. Human ethics Define ‘honest broker’ Animal ethics Acceptable

Further clarification regarding the honest broker system has been added to the Methods section, The Penn State Biorepository staff member who coordinated sample dissemination and patient data collection is a co-author on this manuscript but was not involved in patient sample analysis. The investigators responsible for the experimental analyses were not, and remain not, privy to identifiable patient information, ensuring appropriate separation between clinical data handling and research activities.

The revised manuscript now reads:

Methods Section 2.10: Clinical information for each patient was extracted from the electronic medical record by a neutral system (honest broker) in accordance with the Biorepository protocol to further protect privacy by separating patient identifiers from samples and results.

5. Title: Should the title reflect the importance of sex differences?

We agree with this point that the reviewer has raised, and have edited the title in response:

Systemic iron availability differentially shapes brain and tumor iron handling in a sex-dependent manner in GBM

6. Line 39 Overinterpretation – there is no evidence from clinical data for any association with patient survival

We accept this point from the reviewer and have updated the manuscript to prevent overinterpretation of the clinical results, while still describing the importance.

Abstract: In clinical GBM samples, we observed similar survival trends across varying iron and ferritin levels, suggesting potential translational relevance.

7. Line 56 Oxygen production?

We have updated the manuscript to reflect the correct wording.

Introduction: Iron is an essential element that is required for many physiological processes, including oxygen transport, DNA replication and repair, and neurotransmitter synthesis (5).

8. Line 112 Method clarification needed: What diet was the high-iron group on prior to tumour implant? Were the control and IDA groups maintained on their respective diets (since PND 21) post tumour implant?

The high-iron group was on control diet prior to tumor implant. Control and IDA groups were maintained on their respective diets throughout the study. Clarification has been added to the methods section:

Methods Section 2.1: At the time of tumor implantation, iron-deficient mice continued on iron deficient diets, whereas control diet mice were randomized to either remain on control diets or transition to high iron diets. Diets were maintained for the remainder of the experiment (until the primary endpoint, about 17-37 days post injection), resulting in three experimental groups: control (female n = 8, male n = 9), iron deficient (female n = 8, male n = 9), and high iron (female n = 8, male n= 8; 350 mg/kg Fe, Global rodent diet, 2018 Teklad 18% Protein Rodent Diets, Cat. #NC0035505) diets.

9. Line 175 A loss of 9/50 animals due to monitoring timing seems a waste.

We appreciate the reviewer’s concern regarding animal loss during monitoring. Importantly, animal welfare was prioritized throughout the study. GBM intracranial models are characterized by rapid and unpredictable neurological decline, which can occur in a very short time window during late-stage disease. Although animals were monitored closely in accordance with the IACUC guidelines, several animals experienced such deterioration outside of the scheduled monitoring periods. Animals that progressed rapidly or that could not be reliably perfused were excluded from tissue analyses and accounted for in the reported 9/50. This exclusion was necessary to limit the post-mortem interval. Additionally, this precaution prevented compromised integrity in downstream tissue iron levels from blood pooling or post-mortem iron release.

10. Line 193 Western blot data for FTH1 not shown, why listed here?

This was an error that has been removed from the Methods Section.

11. Line 228, 246 No data on multivariate cox proportional hazards regression is presented. And sentence repeated.

In response to the reviewers, we have replaced the Kaplan-Meier curves in the main figures with the cox proportional hazards regression. The results/figure legends have been updated to represent these results. The Kaplan-Meier curves have been relocated to Supplementary Figures 2-4. We have also deleted the repeat sentence.

12. Fig 3 Why was day 10 chosen? The majority of male mice on control diet did not have measurable tumours. And the majority of female mice on iron deficiency diet similarly showed no tumour. Legend should specify 10 days. No significant difference in survival.

We agree with the reviewer on this point. Day 10 post-tumor injection was selected based on prior literature using this model, which demonstrates reliable tumor establishment ~2mm by 10 days. We intentionally chose an earlier time point both to confirm tumor growth and uptake dynamics prior to the onset of rapid tumor expansion and endpoint-related changes.

As the reviewer notes, a substantial portion of male mice on control diet and female mice on IDA diet did not yet exhibit measurable tumors at day 10. We interpret this as biologically informative, reflecting delayed early growth in these groups. Importantly, this early imaging time point was not intended to predict survival outcome, but the trending relationship between the two was worth mentioning in the manuscript.

The figure legend has been revised to explicitly state that tumor volume was assessed by MRI at 10 days post injection. A brief interpretation of these results has been added to the discussion section:

Discussion: Notably, earlier tumor volume measurement (10 days post injection) in this model provided insight into tumor establishment and iron-dependent growth dynamics. At this stage, several male control and female IDA mice did not yet exhibit measurable tumors, suggesting delayed tumor initiation or early growth restraint in these groups that may have impacted downstream survival outcomes.

13. Fig 4 and 5 There is no evidence from mice that high iron diet resulted in different tumour or brain tissue levels compared to controls. The period of dosing may have been too short. Figure 7 shows blood data – suggest moving closer.

We agree with the reviewer that exposure to high dietary iron did not result in a significant increase in tumor or brain tissue in this model. This finding suggests, and further supports our conclusion, that iron is tightly regulated at the CNS and does not necessarily reflect systemic iron. As the reviewer notes, it is possible that a longer period of exposure may be required to overcome this regulation and induce detectable changes in our outcomes. Our primary rationale for this regimen was to enhance clinical translatability: In the clinical setting, iron supplementation (or dietary iron changes) would almost exclusively occur after diagnosis or in response to anemia identified during treatment rather than prolonged or prior exposure. Therefore, initiating high iron dosing at tumor injection models the earliest realistic intervention point following diagnosis.

While the serum iron data is not statistically significant between controls and high iron groups, high iron males had 68% increase in serum iron compared to controls, while females exhibited a 67% increase. As mentioned, these changes were not complimented by increases in tumor or contralateral hemisphere iron, supporting the conclusion that there are unknown regulatory mechanisms limiting iron uptake at the tissue level despite altered circulating iron.

To improve clarity and better integrate these findings, we have moved the serum iron data to Figure 6, directly following contralateral and tumor iron content.

14. Line 346 A rather optimistic heading as only male mice showed this association; female mice and human patients did not.

We have edited the section title to more accurately reflect these associations:

Section 3.4: Sex-dependent tumor iron content is associated with survival

15. Line 363 There is no trend in clinical data (p=0.31, 0.57 and 0.79).

We agree with the reviewer and have changed the wording to reflect similarities in survival directionality between clinical and preclinical models, while emphasizing that the clinical findings are not statistically significant. These changes also reflect the cox hazards regression.

The revised manuscript now reads:

Section 3.4: When examining clinical data, there is a modest separation in survival curves observed in males, with higher tumor iron associated with slightly longer survival (HR = 0.56 (0.16 – 1.90), p = 0.35), whereas no comparable separation was evident in females (HR = 0.90 (0.29 – 2.77), p = 0.85) or in the combined cohort (HR = 0.98 (0.55 – 1.77), p = 0.96) (Figs 5I-K). Despite the lack of statistical significance in both males and females, the directionality of this pattern was consistent with those observed in the preclinical model.

16. Would it be worth investigating the amount of iron the tumour was able to accumulate in relation to (surrounding/contralateral) normal brain?

We agree with the reviewer that this is an interesting observation. Notably, the tumor has 2x the iron than that of the contralateral hemisphere, which supports the translatability discussed under point #2.

In the present study, contralateral hemispheric iron was used as more of a control to assess brain iron regulation and to evaluate sex-dependent differences at the CNS level. Future studies incorporating spatial resolution or cell-specific changes in iron uptake will be valuable for defining how systemic iron alters the tumor and contralateral hemispheric iron relationship.

17. Tumour contains notably more iron than normal brain. Discuss possible reasons.

This is an important observation by the reviewer and highlights a key translatable component of our model. GBM tumors are highly vascularized, which likely contributes to enhanced iron content specifically through blood components. In addition, GBM cells exhibited elevated transferrin receptor expression and iron demand to support their proliferative and metabolic needs, further contributing to increased iron uptake when compared to surrounding brain tissue. This phenomenon is discussed briefly in the introduction section of the manuscript.

18. Different parts of the brain accumulate different levels of iron; was this considered?

The reviewer makes a good point. In this study, tumors were consistently implanted in the same cortical region, which allowed us to minimize variability related to regional differences in brain iron content. While these regional differences the reviewer mentions are an important area for future investigation, mapping of iron distribution across different brain regions was beyond the scope of our present study.

19. Fig 6 The writing on several graphs is too small to decipher (ie p-values).

Figure 6 has been revised to increase font size 3 points for all p-values

20. Fig 7 No clear difference between control and high iron diet measures.

We agree with the reviewer and have addressed this in point 13. We have also revised the text to explicitly acknowledge the lack of statistical significance yet large change in serum iron values. As previously mentioned, despite these changes in serum iron between control and high iron diet, brain and tumor iron measurements remained unchanged.

Discussion: While the animals were anemic based on hemoglobin and hematocrit measures, serum iron did not differ significantly between control and iron deficient groups or control and high iron groups. It should be noted, however, that high iron males had a 68% increase in serum iron compared to controls, while females exhibited a 67% increase. These increases were not complimented by changes in survival or iron levels in the tumor or brain tissue, further suggesting that systemic iron availability does not directly translate to tissue-specific iron levels.

21. Line 457 There is no trend in clinical data (p=0.75, 0.58, 0.87).

We have updated the manuscript in response to the reviewer to accurately reflect the statistical outcomes.

Section 3.6: However, when males and females were evaluated separately, there were opposing relationships. Male mice with high serum iron had a slight survival advantage over males with low serum iron whereas high serum iron was slightly disadvantageous in females (25 vs 27 days, p = 0.09) (Figs 6F-G), but these trends were not significant in the preclinical model. Similarly, no evidence of a significant survival association with serum iron was observed in the clinical cohort for the combined cohort (HR = 0.7

---

## [Decision Letter · Decision Letter 1]

25 Mar 2026

Dear Dr. Connor,

As the corresponding author, your ORCID iD is verified in the submission system and will appear in the published article. PLOS supports the use of ORCID, and we encourage all coauthors to register for an ORCID iD and use it as well. Please encourage your coauthors to verify their ORCID iD within the submission system before final acceptance, as unverified ORCID iDs will not appear in the published article. *Only* the individual author can complete the verification step; PLOS staff the individual author can complete the verification step; PLOS staff the individual author can complete the verification step; PLOS staff the individual author can complete the verification step; PLOS staff *cannot* verify ORCID iDs on behalf of authors.verify ORCID iDs on behalf of authors.verify ORCID iDs on behalf of authors.verify ORCID iDs on behalf of authors.

We look forward to receiving your revised manuscript.

Kind regards,

Kostas Pantopoulos, PhD

Academic Editor

PLOS One

Journal Requirements:

Reviewers' comments:

Reviewer's Responses to Questions

**Comments to the Author**

Reviewer #1: (No Response)

Reviewer #2: All comments have been addressed

2. Is the manuscript technically sound, and do the data support the conclusions?

Reviewer #1: Partly

Reviewer #2: Partly

3. Has the statistical analysis been performed appropriately and rigorously?

Reviewer #1: Yes

Reviewer #2: Yes

4. Have the authors made all data underlying the findings in their manuscript fully available?

Reviewer #1: Yes

Reviewer #2: Yes

5. Is the manuscript presented in an intelligible fashion and written in standard English?

Reviewer #1: Yes

Reviewer #2: Yes

Reviewer #1: The authors have addressed the majority of my concerns and the manuscript is much improved. However, there is still a tendency to overstate the importance of the clinical data that is clearly not statistically significant, even though it is very carefully worded. I will leave this issue for the editor to decide.

Reviewer #2: Overall, the revised version shows clear improvement compared to the original submission. The authors have addressed most of the reviewers’ concerns, particularly regarding statistical analyses, interpretation of the clinical data, and methodological transparency. The manuscript is now clearer and the conclusions are more balanced.

Although the revised manuscript moderates the interpretation of the clinical cohort throughout the text, the Abstract could benefit from a clearer statement indicating that the associations observed in the clinical dataset did not reach statistical significance. Explicitly stating this point would help ensure that readers do not interpret the clinical findings as definitive evidence but rather as exploratory observations that are consistent with the preclinical trends.

In the same way, the revised manuscript appropriately reduces the level of interpretation applied to the clinical cohort and emphasizes that the observed relationships between circulating iron markers and survival are not statistically significant. This modification strengthens the manuscript. Nevertheless, the authors may consider reinforcing in the Discussion that these findings should be interpreted as exploratory or hypothesis-generating given the limited sample size of the clinical cohort.

.

Reviewer #1: **Yes:** Gabi DachsGabi DachsGabi DachsGabi Dachs

Reviewer #2: **Yes:** Luis IbarraLuis IbarraLuis IbarraLuis Ibarra

---

## [Author Response · Author response to Decision Letter 2]

31 Mar 2026

We have modified several statements in both the abstract and discussion to better reflect the exploratory nature of the clinical analyses. In the abstract, we have added the following sentence: “In clinical GBM samples, we observed non-statistically significant but similar survival trends across varying iron and ferritin levels, suggesting potential translational relevance of our exploratory model.”

We have also revised several sentences in the discussion to improve clarity:

1. Although the clinical analyses did not yield statistically significant associations between serum or tumor iron/ferritin and survival, these exploratory experiments paralleled several key findings in the mouse model, providing important translational context.”

2. “Despite these differences, the preclinical model demonstrated clear sex-dependent effects, with the clinical data showing parallel, but not statistically significant, directionality, highlighting the benefit of integrating both models to generate hypotheses regarding iron regulation in GBM.”

3. “Although the human correlations were not statistically significant in this exploratory study, the consistency of these directional trends support the translational value of the mouse model and implies that the relationships may strengthen as the disease progresses”

---

## [Editor Report · Decision Letter 2]

5 Apr 2026

Systemic Iron Availability Differentially Shapes Tumor and Brain Iron Handling in a Sex-Dependent Manner in Glioblastoma

PONE-D-25-68105R2

Dear Dr. Connor,

We’re pleased to inform you that your manuscript has been judged scientifically suitable for publication and will be formally accepted for publication once it meets all outstanding technical requirements.

Kind regards,

Kostas Pantopoulos, PhD

Academic Editor

PLOS One
---

## [Editor Report · Acceptance letter]

PONE-D-25-68105R2

PLOS One

Dear Dr. Connor,

I'm pleased to inform you that your manuscript has been deemed suitable for publication in PLOS One. Congratulations! Your manuscript is now being handed over to our production team.

Kind regards,

on behalf of

Dr. Kostas Pantopoulos

Academic Editor

PLOS One